

# A New Oxidation Flow Reactor for Measuring Secondary Aerosol Formation of Rapidly Changing Emission Sources

Pauli Simonen[1], Erkka Saukko[1], Panu Karjalainen[1], Hilkka Timonen[2], Matthew Bloss[2], Päivi Aakko-Saksa[3], Topi Rönkkö[1], Jorma Keskinen[1], and Miikka Dal Maso[1]

[1]Aerosol Physics Laboratory, Department of Physics, Tampere University of Technology, P.O. Box 692, FI-33101 Tampere, Finland
[2]Finnish Meteorological Institute, Atmospheric Composition Research, P.O. Box 503, FI-00101 Helsinki, Finland
[3]VTT Technical Research Centre of Finland Ltd., P.O. Box 1000, FI-02044 Espoo, Finland

*Correspondence to*: pauli.simonen@tut.fi

**Abstract**

Oxidation flow reactors or environmental chambers can be used to estimate secondary aerosol formation potential of different emission sources. Emissions from anthropogenic sources, such as vehicles, often vary on short timescales. For example, to identify the vehicle driving conditions that lead to high potential secondary aerosol emissions, rapid oxidation of exhaust is needed. However, the residence times in environmental chambers and in most oxidation flow reactors are too long to study these transient effects. Here, we present a new oxidation flow reactor, TSAR (TUT Secondary Aerosol Reactor), which has a short residence time and near-laminar flow conditions. This allows studying e.g. the effect of vehicle driving conditions on secondary aerosol formation potential of the exhaust. We show that the flow pattern in TSAR is nearly laminar and particle losses are negligible. The secondary organic aerosol (SOA) produced in TSAR has a similar mass spectrum as the SOA produced in the state-of-the-art reactor, PAM (Potential Aerosol Mass). Both reactors produce the same amount of mass, but the TSAR has a higher time-resolution. We also show that the TSAR is capable of measuring secondary aerosol formation potential of a vehicle during a transient driving cycle, and that the fast response of the TSAR reveals how different driving conditions affect the amount of formed secondary aerosol. Thus, the TSAR can be used to study rapidly changing emission sources, especially the vehicular emissions during transient driving.



# 1 Introduction

Aerosol particles in the atmosphere affect climate, health and visibility. To reduce these impacts, the sources of aerosol particles have to be resolved. One large but uncertain source of atmospheric aerosol particles is secondary organic aerosol (SOA) formation, which takes place in the atmosphere when particle mass forms as a result of atmospheric oxidation of organic precursor gases. Because the emission of precursor gases and the formation of secondary aerosol mass occur separately, the estimation of SOA sources and their magnitudes is difficult.

The total amount of atmospheric SOA is typically estimated using laboratory data of SOA yields for known precursors combined with their emission factors and emission profiles. (Kanakidou et al., 2005) However, the uncertainty of this method is high. For example, Kanakidou et al. (2005) estimate that approximately 10 % of global SOA is of anthropogenic origin, but measurements by Volkamer et al. (2006) show that the proportion can be as high as 33 %. Thus, more accurate estimations are needed to identify the most important SOA sources in order to identify the most efficient methods to decrease the human impact on aerosol loading in the atmosphere.

An alternative and more direct method to characterize SOA sources was introduced by Kang et al. (2007). Instead of measuring precursor gases and estimating the amount of potential SOA based on their yields, the SOA formation potential of a single emission source can be measured by oxidizing the emitted sample and measuring the secondary aerosol mass produced. This method reduces the uncertainty of the SOA emission magnitude, since unknown precursors as well as those whose measurement is difficult are taken into account.

Using this in situ method, emission oxidation and SOA formation process can be characterized using large environmental chambers, such as the one Platt et al. (2013) used when they measured the SOA potential of a gasoline vehicle. Another alternative is to use an oxidation flow reactor, in which the sample is oxidized in a similar manner but with higher oxidant concentrations than in large environmental chambers. Such a setup was first introduced by Kang et al. (2007), who also introduced their own oxidation flow reactor, the potential aerosol mass (PAM) chamber, hereafter referred to as PAM. The setup has been used e.g. to estimate the SOA formation potential of in-use vehicle emissions by sampling air from a highway tunnel (Tkacik et al., 2014), to measure the SOA formation from urban ambient air (Ortega et al., 2016) and to measure the SOA formation from ambient pine forest air (Palm et al., 2016). All these applications show the value of direct measurement of SOA potential, since the model results either over- or underestimated the SOA formation.

The use of an oxidation flow reactor instead of a large environmental chamber provides multiple advantages: short residence time, higher degree of oxidation and portability (Bruns et al., 2015). The short residence time allows high time-resolution measurements of constantly changing situations, so for example the effect of different test parameters on SOA formation can





be studied in a shorter time than with environmental chambers. It is also possible to measure SOA formation of a changing emission source in real-time because of the short residence time. For example, Karjalainen et al. (2015) measured time-resolved SOA formation potential of a gasoline vehicle during a transient driving cycle using a PAM reactor. They observed that the secondary aerosol formation potential is highly dependent on the driving conditions. However, the PAM reactor is not ideal

for rapidly changing emission sources such as vehicular emissions, since the residence time is still relatively long and the reactor outputs a distribution of different-aged aerosol (Lambe et al., 2011) .

In this work, we introduce and present a characterization of a new oxidation flow reactor, the *TUT Secondary Aerosol Reactor (TSAR)*. The TSAR is better suited to measuring the real-time secondary aerosol formation potential of rapidly changing

emission sources than the state-of-the-art oxidation flow reactors due to its improved flow conditions and shorter residence time. In the following sections, we characterize the TSAR by describing its particle losses, oxidant exposure, residence time distribution, laboratory studies on sulfur acid yield as well as toluene SOA yield and properties, including a comparison between PAM and TSAR. In addition, we present measurements of the secondary aerosol formation of gasoline vehicle emissions during a transient driving cycle. We show that the fast response of TSAR gives valuable information on the effect

of the driving condition on secondary aerosol formation potential.

## 2 Experimental

### 2.1 Oxidation flow reactor

TSAR is an OFR-254 type oxidation flow reactor, according to terminology proposed by Li et al. (2015), which means that OH radicals are produced from the photolysis of the ozone at 254 nm UV radiation. Its layout is presented in Fig. 1 (see Fig.

S3 for a photograph). The TSAR consists of a residence time chamber (1 in Fig 1), an oxidation reactor (3), an ozone generator, three mass flow controllers and an expansion tube (2) that connects the residence time chamber and oxidation reactor. The residence time chamber is a 50 cm x 5 cm ID stainless steel cylinder that ensures the mixing of the sample and makes the sample flow laminar before entering the oxidation reactor. The half cone angle of the expansion tube is 6 degrees. Two of the mass flow controllers are connected to a vacuum line and are used to control the flow rates inside the residence time chamber

and the oxidation reactor. The excess flow from the oxidation reactor is hereafter called "secondary excess flow". The third mass flow controller adjusts the air flow through the ozone generator. All the components except the residence time chamber and the expansion tube are located inside a single housing, which makes TSAR easy to transfer to different measuring environments.

The TSAR oxidation reactor is a 3.3 l (52 cm x 9 cm inner diameter) quartz glass cylinder surrounded by two ozone free low pressure mercury lamps which emit 254 nm UV light. The lamps are placed outside the reactor to ensure laminar flow and to decrease the surface-to-volume ratio. The UV radiation generates excited oxygen atoms [O(1D)] from the photolysis of $O_3$.



These atoms react with water molecules, producing OH radicals. The $O_3$ needed for this reaction chain is mixed with the sample prior to the residence time chamber. In some cases, the humidity of the sample is too low for sufficient OH generation and additional humidification is required; in these cases, humidified air is also mixed into the sample at this point.

The ozone is generated by an external ozone generator (either Model 600 or Model 1000, Jelight Company Inc.), which produces ozone from oxygen photolysis by 185 nm UV radiation. The ozone concentration can be adjusted by partially covering the UV lamp (Model 600) or by adjusting the flow rate through the generator.

The TSAR outlet is a 10 mm OD stainless steel probe, and its axial position can be adjusted, so that the oxidized sample can

be measured from any distance from the inlet. From the probe, the sample is led to the measurement devices or to an ejector diluter, which allows the use of multiple instruments while maintaining a constant flow through the oxidation reactor.

**2.1 Residence time distribution experiments**

The flow conditions inside the TSAR oxidation reactor affect the dynamic transfer function $E(t)$ of the reactor for non-reacting compounds. For this case, the measured temporal output concentration $C_{out}(t)$ of the TSAR for a measured dynamic input

concentration $C_{in}(t)$ is the convolution of the measured input concentration and the transfer function (Fogler, 2006):

$$C_{out}(t) = E(t) * C_{in}(t) \tag{1}$$

The transfer function $E(t)$ also is the unit impulse response of the reactor or the residence time distribution following an ideal Dirac delta input impulse. To test the response function, ten-second square pulses of $CO_2$ were injected into the TSAR mixed

with pressurized air. To keep the shape of the $CO_2$ pulse as sharp as possible, the volumetric flow rate in the residence time chamber was kept at 50 slpm. In the oxidation reactor the flow rate was 5 slpm. $CO_2$ concentration was measured with a $CO_2$ analyzer (Sidor, Sick Maihak). As the same instrument is used for the measurement of both input and output concentrations, its response function is imbedded both in $C_{in}(t)$ and $C_{out}(t)$.

First, three separate $CO_2$ pulses were measured with sampling at the end of the residence time chamber. The outlet probe was then adjusted to sample at the end of the oxidation reactor and three separate pulses were again measured. The residence time distributions were determined for different situations: UV lamps and the secondary excess flow were either on or off.

**2.2 Particle loss quantification**

Particle losses in the oxidation reactor were measured using dioctyl sebacate (DOS) particles with mobility diameter from 20

to 100 nm and silver particles from 5 to 30 nm. The DOS particles were generated by atomizing DOS-isopropanol solution. The silver particles were generated with an evaporation-condensation technique (Harra et al., 2012). In these experiments, the volumetric flow in both the residence time chamber and oxidation reactor was 5 slpm.





A narrow monodisperse particle size distribution, size-selected using a differential mobility analyzer (nano-DMA, TSI Inc. Model 3085), was injected into the TSAR. The particle number concentration was measured with an ultrafine condensation particle counter (UCPC, TSI Inc. Model 3025) before and after the oxidation reactor using the adjustable outlet probe. This

procedure was repeated two or three times for each particle size.

**2.4 OH exposure experiments**

The length of the duration of atmospheric oxidation that the oxidation flow reactor simulates is determined by exposure of the sample to OH radicals. OH exposure ($OH_{exp}$) is defined as $[OH] \times t$, where $[OH]$ is the mean OH radical concentration in the oxidation reactor and $t$ is the mean residence time of sample in the reactor. $OH_{exp}$ could be measured indirectly by monitoring

the loss of $SO_2$ in the reactor.(Lambe et al., 2011) Since the only significant loss of $SO_2$ in the oxidation reactor is due to the reaction with OH radicals, the change in $SO_2$ concentration is defined by the following differential equation:

$$\frac{d[SO_2]}{dt} = -k_{OH+SO_2}[OH][SO_2],\qquad(2)$$

where $[SO_2]$ is the $SO_2$ concentration and $k_{OH+SO_2}$ is the reaction rate constant. From this, we get the OH exposure

$$OH_{exp} = \frac{1}{k_{OH+SO_2}}\ln\frac{[SO_2]_0}{[SO_2]_f},\qquad(3)$$

where $[SO_2]_0$ and $[SO_2]_f$ are the $SO_2$ concentrations of the sample before and after oxidation, respectively.

Because the OH radicals are produced in a reaction between water molecules and $O(^1D)$ atoms produced by ozone photolysis, both humidity and ozone concentration affect the amount of OH radicals (Seinfeld and Pandis, 1998). $OH_{exp}$ was measured using three different relative humidities (15 %, 30 % and 45 %) and several different ozone concentrations (0.6 ppm–49 ppm).

Pressurized air, $SO_2$, humidified air and ozone were injected into the TSAR to determine the OH exposure. First, humidity, ozone concentration and $[SO_2]_0$ were measured after the TSAR. Then the UV lamps were turned on, and the concentration rapidly decreased and stabilized to the value of $[SO_2]_f$. $SO_2$ concentration was measured with AF22M analyzer (Environnement S.A) and ozone with model 205 analyzer (2B Technologies).

**2.5 Sulfuric acid yield experiments**

$SO_2$ oxidation is a simple example of secondary aerosol formation. $SO_2$ reacts with OH radicals to produce sulfuric acid ($H_2SO_4$) vapor which rapidly enters the particle phase by nucleation and condensation (Sihto et al., 2006). The mass formed by oxidation of $SO_2$ can be theoretically calculated from the $SO_2$ loss, and thus comparing the measured mass formation to the theoretical prediction can be used to estimate the capability of TSAR to simulate full atmospheric oxidation. Should the measured mass be substantially smaller than the theoretical, we would assume that there are significant losses of sulfuric acid



vapor inside TSAR. This would then also mean that the losses for organic low-volatile and semi-volatile vapors were also high.

The sulfuric acid yield was measured by injecting pressurized air, SO$_2$, humidified air and ozone into the TSAR. The relative humidity and SO$_2$ was measured straight after TSAR, whereas ozone concentration and the particle size distribution were measured after an ejector dilutor (Dekati Ltd.). The dilution ratio was determined by measuring the sample flow rate and the dilution air flow rate. The particle size distribution was measured with a nano-SMPS (scanning mobility particle sizer: a nano-DMA (TSI Inc. model 3085) combined with a UCPC (TSI Inc. model 3025)).

In addition to sulfuric acid, the measured particles also contain water. The sulfuric acid mass was calculated from Eq. (4) (Lambe et al., 2011).

$$m_{H_2SO_4} = \chi_{H_2SO_4} \times V \times \rho, \tag{4}$$

where $\chi_{H_2SO_4}$ is the mass fraction of sulfuric acid in the particle phase, $V$ is the volume calculated from nano-SMPS particle size distribution and $\rho$ is the density of the particle phase. Both the mass fraction and the density were calculated as a function of relative humidity based on Seinfeld and Pandis (1998). In the calculations, relative humidity after the dilution is used, assuming fast equilibration of the sulfuric acid particles.

The theoretical (maximum) sulfuric acid mass was calculated by multiplying the loss of SO$_2$ by the molar mass of a sulfuric acid molecule. Thus, the loss of 1 ppb of SO$_2$ produces 4.03 µg m$^{-3}$ of sulfuric acid aerosol, assuming also that all the sulfuric acid condenses into the particle phase.

**2.6 Organic precursor experiments**

A key application of TSAR is to estimate the amount of secondary aerosol mass formed from engine exhaust emissions, which in turn contains a complex mixture of organic and inorganic gases. Therefore, the SO$_2$ oxidation experiment alone is not a representative example of engine exhaust oxidation, because the oxidation pathways of organic compounds are far more complex. The ability of TSAR to form SOA was verified by measuring the toluene SOA obtained by TSAR and the PAM simultaneously. Previous studies have shown that the amount and properties of the SOA produced in the PAM are similar to those of the SOA formed in smog chambers, which represent atmospheric oxidation (Bruns et al., 2015; Lambe et al., 2015).

The organic precursor gas in this experiment was toluene, because it is present in engine exhaust gas (Peng et al., 2012; Wang et al., 2013). In addition, toluene is globally one of the most emitted anthropogenic SOA precursors (Kanakidou et al., 2005) Gas-phase toluene was produced using a permeation oven with a toluene permeation tube (KIN-TEK Laboratories Inc.), and its output rate ($\dot{M}_{toluene}$) was measured by weighing the change in its mass. The concentration of toluene in the reactors is



$$C_{toluene} = \frac{\dot{M}_{toluene}}{Q_{tot}}, \tag{5}$$

where $Q_{tot}$ is the total sample flow through the reactors (10 slpm).

The gas-phase toluene was mixed with ozone and humidified air before it was fed to the TSAR residence time chamber. After
the residence time chamber, 5 slpm of the sample was introduced into the TSAR oxidation reactor and 5 slpm to the PAM. A
4-way valve was installed after the reactors, so that the instruments were sampling from one reactor while the sample from the
other reactor was drawn to the vacuum line through a mass flow controller.

The PAM was used in OFR185 mode (Li et al., 2015) and thus the external ozone generator was switched off when the
instruments were sampling from the PAM. The $OH_{exp}$ of the reactors was varied by varying the light intensity in the PAM and
the amount of injected ozone in the TSAR. The PAM $OH_{exp}$ as a function of output ozone concentration was measured off-
line in a similar way as for TSAR (Sect. 2.4) at 28 % relative humidity. The PAM reactor $OH_{exp}$ as a function of output ozone
concentration is shown in Fig. S1.

The particle size distribution downstream of TSAR and PAM was measured with an SMPS (Model 3081 DMA and Model
3775 CPC, TSI Inc.) and in some experiments also with an engine exhaust particle sizer (EEPS, TSI Inc.) (Johnson et al.,
2004). The EEPS sample had to be diluted with a mass flow controller to keep the total flow rate through chambers at 5 slpm.
Aerosol chemical composition and size distribution were measured with an SP-AMS (Soot-particle aerosol mass spectrometer,
Onasch et al., 2012). In addition, the ozone concentration (Model 205, 2B Technologies) and relative humidity (Hygroclip
SC05, Rotronic AG) were measured.

Two different toluene experiments were run: steady state and pulse experiments. In the steady state experiments, a constant
concentration of toluene was continuously injected into the reactors. Based on these experiments, the toluene SOA yield was
determined for both reactors.

The pulse experiments were performed to study the reactors' behavior during rapid changes of toluene concentration. In these
experiments, toluene was injected through a 3-way solenoid valve to either the reactors, or to the excess line. Three different
pulse experiments were performed: a single 10 second pulse, and two different cycles with several pulses (cycle 1 and cycle
2). In cycle 1, three toluene pulses were injected with intervals of 10 and 15 seconds, whereas cycle 2 had intervals of 40 and
50 seconds. The cycles are described in detail in Table 1.

In both cycles, the total toluene injection time was 25 seconds and therefore the total amount of injected toluene was equal.
The EEPS was used to measure the particle number distribution of produced SOA at a time resolution of one second. For the





pulse experiments, the flow rate through each reactor was 5 slpm. Since the PAM is approximately twice as big as TSAR in volume, also a 10 slpm flow rate was used for the PAM to compare the reactors at similar theoretical residence times. In this case, the TSAR was bypassed to keep the total flow at 10 slpm.

The SOA yield ($Y$) is defined as the produced organic aerosol mass ($\Delta M$) per reacted precursor mass ($\Delta HC$) (Odum et al., 1996):

$$Y = \frac{\Delta M}{\Delta HC}. \tag{6}$$

The amount of reacted toluene mass depends on the $OH_{exp}$; the change in toluene concentration is defined by a similar differential equation as the change in $SO_2$ concentration (Eq. (2)). Thus, the amount of reacted toluene is

$$\Delta[toluene] = [toluene]_0(1 - \exp(-k_{OH+toluene} \times OHexp)), \tag{7}$$

where $[toluene]_0$ is the initial toluene concentration and $k_{OH+toluene}$ is the reaction rate constant between toluene and OH radicals. A rate constant of $6.18 \times 10^{-12}$ cm$^3$ s$^{-1}$ was used based on the parameters presented by Atkinson (1985).

**2.7 Vehicle exhaust experiments**

The ability of TSAR to produce secondary aerosol mass from engine exhaust emissions was evaluated by sampling the exhaust
of a Euro 5 GDI light-duty vehicle during a transient driving cycle (New European Driving Cycle, NEDC) run on a chassis dynamometer. The official cycle begins with a cold engine start, but in this study the NEDC was run with a warm engine and this is hereafter called a *warm NEDC*. Prior to the warm NEDC, the vehicle was run at 80 km h$^{-1}$ for at least 3 minutes, and the cycle began with an idling engine.

The sampling setup of vehicle exhaust experiments is shown in Fig. S2. The engine exhaust was sampled from the tailpipe using a porous-tube diluter (PTD) followed by a short cylindrical residence time chamber with residence time of 2.9 s. Dilution air temperature was 30 °C and dilution ratio approximately 12. This dilution setup has been shown to mimic the atmospheric cooling and dilution processes of primary aerosol reasonably well (Keskinen and Rönkkö, 2010; Rönkkö et al., 2006). The exact dilution ratio of the PTD was determined by $CO_2$ measurements from the tailpipe and after the PTD. After the residence
time chamber, 3 slpm of humidified air and 3 slpm of ozone were mixed with the sample. At this stage, the dilution ratio was 2.5. Thus, the total dilution ratio before TSAR was approximately 30. The sample from TSAR was drawn through an active carbon ozone scrubber to an ejector diluter (Dekati Ltd.) at 5 slpm flow rate. The total dilution ratio between the tailpipe and instruments was determined by $CO_2$ measurements which were performed during 80 km h$^{-1}$ steady state driving, when the $CO_2$ concentration in the tailpipe was stable.


The particle size distributions were measured with EEPS, Electrical low-pressure impactor (ELPI+, Dekati Ltd.) and High-resolution low-pressure cascade impactor (HRLPI)(Arffman et al., 2014). $CO_2$ concentration after tailpipe was measured with





Sick Maihak $CO_2$ analyzer, using a sample drier prior to the analyzer. Relative humidity and sample temperature were measured after TSAR using an RH sensor (Hygroclip SC05, Rotronic AG).

The amount of secondary aerosol mass produced in TSAR was determined by subtracting the primary mass from the mass measured when using the TSAR. Primary aerosol was measured with the same setup by operating the TSAR with UV lamps and the ozone generator turned off. The primary emission was measured during two warm NEDCs.

In this setup, the sample flow from the tailpipe is constant regardless of the exhaust mass flow. To determine the emission factors, the measured concentrations are multiplied with the corresponding exhaust mass flow.

## 3 Results and discussion

### 3.1 Residence time distribution

The evolution of a $CO_2$ pulse in the TSAR is shown in Fig. 2. A narrow pulse enters the oxidation reactor and exits the reactor as a broader pulse. The theoretical transfer function of the oxidation reactor is calculated based on the residence time distribution of ideal laminar flow:

$$E(t) = \begin{cases} 0 & , t < \frac{\tau}{2} \\ \frac{\tau^2}{2t^3} & , t \geq \frac{\tau}{2} \end{cases}, \tag{8}$$

where the constant $\tau$ is defined as

$$\tau = \frac{\pi R^2 L}{Q}, \tag{9}$$

where $R$ is the inner radius of the reactor, $L$ is the length of the reactor and $Q$ is the flow rate. (Fogler, 2006)

Figure 2 shows both the measured pulse after the reactor and the modeled pulse calculated according to Eq. (1) using the theoretical transfer function and the measured input concentration.

As seen in Fig. 2, the measured pulse is somewhat broader than the modeled one. There are some possible reasons for this discrepancy: first, the flow inside the reactor is probably not totally laminar because of the expansion in diameter between the residence time chamber and the oxidation reactor, and because of the abrupt diameter change at the end of the reactor; second, the pulse becomes broader in the sampling lines, which is not taken into account here.

In Fig. 2, UV lamps are turned off and the secondary excess flow is on. Because both of these affect the flow, the residence time distribution was measured for different combinations of these parameters, and the results are shown in Fig. 3. In all cases, the total flow rate through the oxidation reactor was 5 slpm. Because the incoming pulse is not an ideal Dirac delta function,





the residence time distribution cannot be calculated with Eq. (8). Instead, the residence time distribution is the measured concentration $C_{out}$(t) divided by the total area of the pulse (Fogler, 2006)

$$RTD(t) = \frac{C_{out}(t)}{\int_0^\infty C_{out}(t)dt} .$$  (10)

Turning off the secondary excess flow broadens the distribution slightly, probably because there is more dead volume at the end of the reactor. Turning the UV lamps on has a similar effect. The UV lamps heat the reactor walls and cause convection inside the reactor. This effect could be reduced by circulating air through the TSAR housing, but on the other hand small heating of the reactor walls may decrease the vapor wall losses. Another method to reduce the convection is to place the reactor vertically.

The residence time distributions show that the flow in the TSAR oxidation reactor is near-laminar. Thus, the mean residence time of the sample in the reactor can be calculated with Eq. (10) (Fogler, 2006)

$$t_{mean} = \frac{\tau}{2} = \frac{\pi R^2 L}{2Q},$$  (11)

which yields 37 seconds at 5 slpm flow rate. Turning off the secondary excess flow reduces the laminarity, but this is often
necessary to keep the flow rate at 5 slpm since for example an ejector diluter alone draws approximately 5 slpm of sample. In any case, the residence time distribution is clearly narrower, and the mean residence time is shorter than those of the PAM (Lambe et al., 2011), allowing the measurements of rapidly changing emission sources.

### 3.2 Particle losses

The particle transmission efficiency as a function of particle mobility diameter is presented in Fig. 4, as well as the theoretical
diffusive losses of particles in a tube with laminar flow (Brockmann, 2011). The markers indicate the particle material, and error bars denote the standard deviation between separate experiments.

Figure 4 shows that the measured transmission efficiency agrees well with the theoretical efficiency, as expected, and thus the losses are less than 10 % when the particle mobility diameter is larger than 5 nm. Therefore, the results in the next sections are
not corrected with this efficiency curve because the particle losses are negligible.

According to Lambe et al. (2011), the transmission efficiency of particles is significantly lower in the PAM: less than 70 % for particles smaller than 100 nm. Since the flow in TSAR is near-laminar, it is not surprising that the measurements agree with the theory. In PAM, the residence time distribution is broad, allowing more time for the particles to diffuse onto walls
(and possibly to coagulate or evaporate), resulting in a non-ideal transmission efficiency.



### 3.3 OH exposure

Figure 5 shows that the OH exposure in TSAR oxidation reactor is sensitive to ozone concentration at low concentrations, but levels off to a near-constant value when the concentration is higher than 25 ppm. The $OH_{exp}$ also depends on the relative humidity. The maximum $OH_{exp}$ at 30 % RH is approximately $1.2 \times 10^{12}$ molec. s cm$^{-3}$, equivalent of 9 days of atmospheric OH exposure, whereas in the PAM, up to 17 days of equivalent exposure are reached (Lambe et al., 2015; See also Fig. S1). In the calculation of the equivalent atmospheric exposure, an average OH concentration of $1.5 \times 10^6$ molec. cm$^{-3}$ in the atmosphere is assumed (Mao et al., 2009). TSAR OH exposure could be further increased by increasing the RH or by increasing the UV lamp wattage.

### 3.4 Sulfuric acid yield

The measured sulfuric acid mass as a function of expected mass is shown in Fig. 6. For the three measurements with the smallest error bars, the measured mass is on average 4 % lower than the expected mass, indicating that there are no significant losses of sulfuric acid vapor or particles.

### 3.5 Toluene SOA yield and properties

Toluene concentration in the sample entering the reactors was 320 ppb ($\pm$34 ppb). The temperature of sample was 23 – 24 °C and average relative humidity 26 – 34 %, with the exception of the PAM measurement at highest OH exposure, where the average relative humidity was 37 %.

#### 3.5.1 Steady state experiments

The SOA mass formed in the reactors is calculated from the number size distribution measured by the SMPS, assuming spherical particles with a density of 1.45 g cm$^{-3}$ (Ng et al. 2007). The SMPS was used for PM concentration measurements instead of the AMS because especially for the TSAR, all particles do not fall in the AMS detection range (40 nm–800 nm). The background mass, i.e. the mass formed in the reactors in the absence of toluene, was subtracted from the toluene SOA mass.

Figure 7 shows SMPS mass distributions of toluene SOA for PAM and TSAR at $OH_{exp}$ of $1.3 \times 10^{12}$ molec. s cm$^{-3}$ and $1.1 \times 10^{12}$ molec. s cm$^{-3}$, respectively. The PAM produces a wide mass distribution where particles above 100 nm contribute to approximately half of the total mass. The TSAR produces a narrower mass distribution where approximately half of total mass is located in particles smaller than 40 nm. This phenomenon was also reported by Bruns et al. (2015): the micro-smog chamber, which is smaller, has a shorter residence time, and generates smaller particles than PAM.





Figure 8 shows toluene SOA yield obtained in steady state experiments as a function of $OH_{exp}$ for both reactors. The $OH_{exp}$ was not measured simultaneously but is obtained from the off-line calibrations. The maximum yield is approximately 0.2 for both reactors (neglecting the TSAR outlier at $OH_{exp}$ of $1 \times 10^{12}$ molec. s cm$^{-3}$), and both reactors reach the maximum yield at $OH_{exp}$ between $0.5 \times 10^{12}$ molec. s cm$^{-3}$ and $1.5 \times 10^{12}$ molec. s cm$^{-3}$. However, the TSAR yield is more sensitive to $OH_{exp}$

than the PAM yield. One possible reason is that because of the broad residence time distribution, the PAM always outputs SOA with both high and low $OH_{exp}$ regardless of the mean $OH_{exp}$. This mixing of different-aged SOA may stabilize the output and reduce the sensitivity to $OH_{exp}$.

Kang et al. (2007) reported a toluene SOA yield of 0.09 using an early version of the PAM at similar relative humidity,

temperature and toluene concentration as in this paper. Ng et al. (2007) and Hildebrandt et al. (2009) used smog chambers with seed particles and low relative humidity, and reported toluene SOA yields of 0.30 and 0.26-0.41, respectively. The SOA yield in this study is therefore slightly lower than in smog chamber studies, which may be caused by the lack of seed particles: Lambe et al. (2015) showed that the addition of seed particles in the PAM resulted in a higher SOA yield, at least in the case of isoprene.

In addition to yield, the chemical composition of produced SOA was studied. In Fig. 9, a van Krevelen diagram shows the oxidation state of SOA for both reactors. In this diagram, H/C ratio is shown as a function of O/C ratio. Elemental ratios are calculated using the method developed by Aiken et al. (2008). Oxidation of aerosol usually increases the O/C ratio and decreases the H/C ratio (Heald et al., 2010). This phenomenon is observed in both reactors, and based on these ratios, the

oxidation state of SOA is similar in PAM and TSAR at comparable OH exposures when the exposure is less than 7 equivalent days. At higher $OH_{exp}$, similar ratios are still observed in both reactors, but at different OH exposures.

To further compare the SOA oxidation state in the reactors, the average carbon oxidation state ($\overline{OS_C}$) of SOA is shown in Fig. 10. The average carbon oxidation state is a metric which is invariant to hydration or dehydration, and is defined as $\overline{OS_C} \approx 2 \times$

$O/C - H/C$ (Canagaratna et al., 2015; Kroll et al., 2011). As well as the O/C ratios and H/C ratios, the $\overline{OS_C}$ of the SOA in the reactors also differ at high $OH_{exp}$. The TSAR SOA has higher oxidation state than the PAM SOA even though the $OH_{exp}$ is similar.

These discrepancies in the oxidation state suggest that the $OH_{exp}$ alone does not affect the composition of SOA. For example,

in Fig. 10 the $\overline{OS_C}$ of TSAR SOA increases from 0.32 to 0.63 even though the increase in $OH_{exp}$ is small ($2.0 \times 10^{11}$ molec. s cm$^{-3}$). PAM SOA needs an increase of $4.0 \times 10^{11}$ molec. s cm$^{-3}$ in $OH_{exp}$ to achieve the same increase in $\overline{OS_C}$ (from 0.34 to 0.63). The difference between TSAR and PAM in the high $\overline{OS_C}$ region is that the outlet ozone concentration in TSAR increases by 42 ppm when $OH_{exp}$ increases by $5.1 \times 10^{11}$ molec. s cm$^{-3}$, whereas in PAM the outlet ozone concentration increases only




by 7.2 ppm at comparable change in $OH_{exp}$ ($6.4 \times 10^{11}$ molec. s cm$^{-3}$). Thus, the increasing ozone exposure in TSAR may lead to the increase in $\overline{OS_C}$, which suggests that the toluene oxidation products react with ozone. However, also the $OH_{exp}$ estimation may be uncertain, since it is based on off-line calibration. In this experiment, the high concentration of toluene likely decreases the $OH_{exp}$, and this effect may have a different magnitude between the reactors. This could also explain the

discrepancies in Fig. 10.

The chemical composition of SOA is further compared by studying the organic mass spectra. According to Marcolli et al. (2006) and Lambe et al. (2015), a dot product between two normalized mass spectra can be used to determine whether the spectra are similar. The spectra are normalized by dividing each signal by the square root of the sum of the squares of all

signals. A dot product of one implicates that the spectra are identical, and zero that they are orthogonal.

Toluene SOA here is divided in three categories: low oxidation ($-0.18 < \overline{OS_C} < -0.16$), medium oxidation ($0.50 < \overline{OS_C} < 0.69$) and high oxidation ($\overline{OS_C} > 1.10$). The dot products between the organic spectra of different reactors are shown in Table 2. The dot products of normalized mass spectra of SOA produced in reactors at comparable $\overline{OS_C}$ are above 0.99, indicating that the

reactors produce similar SOA matter in regard to chemical composition.

The TSAR and PAM reactors differ in volume, geometry, flow conditions and residence time. The most significant difference is in the oxidation process: TSAR operates in OFR254 mode and PAM in OFR185 mode. However, the agreement between yields and organic mass spectra of SOA produced in both the TSAR and PAM reactors show that the oxidation products are

similar in both reactors, at least in the case of toluene. In OFR254, the sample is first exposed to ozone (before the oxidation reactor) and then to both ozone and OH radicals. If the VOCs in the sample react fast with ozone, resulting SOA mass might differ between OFR254 and OFR185. This was not the case for toluene, as dark experiments (only ozone, no UV light) did not produce any secondary mass.

### 3.5.2 Pulse experiments

The SOA mass concentrations as a function of time are shown in Fig. 11 for all pulse experiments. The 10 s pulse of toluene results in a sharp peak in mass in the TSAR, whereas the PAM reactor produces significantly broader peaks at both used flow rates. Interestingly, the TSAR mass peak is divided into two distinct peaks. We do not know the reason for this phenomenon since the residence time distributions in Sect 3.1 do not support this kind of behavior. However, the flow conditions in this experiment are not exactly the same as in Sect. 3.1: here, the flow rate in residence time chamber is only 10 slpm, whereas in

Sect. 3.1 it was 50 slpm.

Cycle 1 with three rapid toluene pulses shows the importance of laminar flow and short residence time in TSAR: PAM produces only one broad peak whereas all the three pulses can be distinguished in the SOA mass produced by the TSAR. In



cycle 2, where toluene pulses are injected between longer intervals, the pulses are also separated in the mass produced by PAM.

As the total amount of toluene injected into the reactors is known and the yield is determined in Sect. 3.5.1, the total mass produced in the reactors can be predicted with Eq. (12).

$$M_{predicted} = M_{tol} \times Y, \tag{12}$$

where $M_{tol}$ is the total mass of the toluene injected and $Y$ is the yield. In these experiments, the $OH_{exp}$ was approximately $0.9 \times 10^{12}$ molec. s cm$^{-3}$ in TSAR and $1.3 \times 10^{12}$ molec. s cm$^{-3}$ in PAM, so a yield of 0.2 is used for both reactors to calculate the expected mass. The mass produced in the reactors is the area of the peaks in Fig. 10 multiplied by the flow rate through the reactors. The comparison between the expected mass and the formed mass is shown in Fig. 12.

In all the experiments, the mass produced in the reactors agree well with the expected mass. For the 10 s pulse, PAM mass is lower than the expected mass and TSAR mass; for cycle 1, TSAR produces more mass than expected; and for cycle 2, both reactors produce less mass than predicted. Considering the uncertainties in this experiment, namely the dilution ratio, EEPS inversion and toluene concentration, we conclude there are no significant differences in the total mass the reactors produce, even though the pulse shapes are clearly different. In all the cases, the mass produced in PAM is slightly lower than in TSAR, probably because the particle losses in PAM are higher.

The agreement between the predicted mass and the produced mass suggest that the approach to measure the secondary aerosol formation potential in real-time is valid: the narrow residence time distribution of TSAR gives time-resolved information of SOA formation from fast changing precursor concentrations but still produces approximately the same amount of mass as the PAM reactor, where the oxidation process is slower.

### 3.6 Engine exhaust oxidation

When measuring time-resolved secondary aerosol formation during a transient driving cycle, it is crucial to synchronize real-time aerosol measurements with vehicle speed data. This is performed by comparing the $CO_2$ measurements in tailpipe and after the dilution steps.

### 3.6.1 TSAR oxidation

In Sect. 3.3 we showed that the $OH_{exp}$ in TSAR depends on relative humidity and ozone concentration. In engine exhaust experiments RH was 33-36 % and temperature 22 °C. Ozone concentration was not measured but based on later laboratory experiments, the ozone concentration has been approximately 11 ppm in the sample flow. According to the results presented in Sect. 3.3, the $OH_{exp}$ with this ozone concentration is approximately $8 \times 10^{11}$ molec. s cm$^{-3}$ (equivalent photochemical age



of 6.3 days). However, this should be considered only as an upper limit for the $OH_{exp}$. There is always NO present in the exhaust sample, and the ozone reacts fast with NO, titrating practically all NO to $NO_2$ before the sample enters the TSAR. Therefore, NO emissions cause a loss in ozone concentration, suppressing the $OH_{exp}$. In addition, $NO_2$ and other OH reactive compounds in the exhaust further decrease the $OH_{exp}$. As the concentrations of $NO_x$ and other gaseous compounds vary during

the driving cycle, so does the $OH_{exp}$. The time resolved $OH_{exp}$ during the driving cycle should be determined by monitoring an OH reactive tracer, such as CO. In this work, the $OH_{exp}$ was not measured in real-time.

### 3.6.2 Time-resolved secondary aerosol formation

The secondary aerosol mass concentration formed from the GDI exhaust during a warm NEDC is shown in Fig. 13. Mass concentration is calculated from particle number size distribution measured by EEPS assuming spherical particles with a

density of 1.0 g $cm^{-3}$ and multiplying the result by the total dilution ratio. The shown mass concentration is an average value of two identical warm NEDCs, and the standard deviation between these two measurements is shown as shaded area. A constant value of background mass formed from dilution air has been subtracted from the calculated mass.

Figure 13 shows significant differences in secondary aerosol formation during different driving conditions. The small standard

deviation suggests that the operation of TSAR and the phenomena causing secondary aerosol formation are highly reproducible. The least secondary aerosol formation occurs during long steady state driving, such as the one at 70 km $h^{-1}$ at the end of the cycle. When the car is accelerated to 100 km $h^{-1}$ and to 120 km $h^{-1}$ at the end of the cycle, the secondary aerosol mass formation increases.

We also observe a new phenomenon, where engine braking results in high concentrations of secondary aerosol forming precursors. Every deceleration (i.e. engine braking) during warm NEDC produces a peak in secondary mass concentration. The tail at the beginning of the cycle is also a result from engine braking, as steady-state driving at 80 km $h^{-1}$ was always performed before the warm NEDC.

The time-resolved emission factor of secondary aerosol mass in Fig. 13 is achieved by multiplying the secondary mass concentration by the exhaust mass flow. Low exhaust mass flow during engine braking cancels out the high mass concentration peaks. Instead, the peak at the end of the cycle dominates the emissions of secondary aerosol precursors.

Since no aerosol chemical composition measurements were performed, we cannot specify the amount of organic mass in the

formed secondary aerosol, and therefore we do not present the emission factor for SOA potential of the engine exhaust. In addition, the high background mass (i.e. unclean dilution air) and the lack of real-time $OH_{exp}$ measurement make this data qualitative rather than quantitative. However, this experiment shows the feasibility of TSAR to measure the time-resolved secondary aerosol formation potential of rapidly changing vehicle emissions. This way, we can identify the driving conditions



in which most secondary aerosol forming precursors are emitted. If the sample was injected to a smog chamber with a constant dilution ratio and then oxidized, like Platt et al. (2013) did, the precursor pulses emitted during engine braking events would cause an over-estimation of total secondary aerosol formation potential.

## 4 Conclusions

In this work, we introduced TSAR, a new short-residence-time oxidation flow reactor for secondary aerosol formation studies. We studied the performance of the reactor by measuring the sulfuric acid yield, toluene SOA yield as well as the composition and the secondary aerosol formation potential of light-duty gasoline vehicle exhaust during a transient driving cycle. In addition, we characterized the particle transmission efficiency and the residence time distribution of the reactor.

The toluene experiments show that both SOA yield and composition are similar in TSAR SOA as in PAM SOA, even though PAM operates in OFR185 mode and TSAR in OFR254 mode. The similarity indicates that TSAR is able to simulate atmospheric SOA formation since Bruns et al. (2015) and Lambe et al. (2015) show that the composition of SOA formed in PAM is similar to the SOA formed in a smog chamber.

The particle losses in TSAR are negligible and the flow is near-laminar. These properties, together with short residence time, make TSAR better suited for monitoring the secondary aerosol formation potential of rapidly changing emission sources than the PAM chamber. We demonstrate the importance of this feature by measuring the secondary aerosol formation of car exhaust during a driving cycle. This experiment shows that TSAR is able to differentiate which driving conditions are most significant regarding the secondary aerosol formation potential.

## 5 Data availability

The data of this study is available from the authors upon request.

## 6 Acknowledgements

The TSAR was designed and built in the project "Finnish-Chinese Green ICT R&D&I Living Lab for Energy Efficient, Clean and Safe Environments", financially supported by Finnish Funding Agency for Innovation (Tekes), and Ahlstrom Oy, FIAC Invest Oy, Green Net Finland Oy, Kauriala Oy, Lassila & Tikanoja Oyj, Lifa Air Oy, MX Electrix Oy, Pegasor Oy and Sandbox Oy.





The TSAR characterization was conducted in the framework of the HERE project funded by Tekes (the Finnish Funding Agency for Innovation), Agco Power Oy, Dinex Ecocat Oy, Dekati Oy, Neste Oyj, Pegasor Oy and Wärtsilä Finland Oy.

Pauli Simonen acknowledges Tampere University of Technology Graduate School.

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

**Table 1: Toluene injection cycles**

| Cycle 1 | | | Cycle 2 | |
|---|---|---|---|---|
| Time (s) | Injection | | Time (s) | Injection |
| 0 | on | | 0 | on |
| 10 | off | | 5 | off |
| 20 | on | | 45 | on |
| 25 | off | | 55 | off |
| 40 | on | | 105 | on |
| 50 | off | | 115 | off |

**Table 2. Dot product between the organic spectra of PAM and TSAR generated SOA.**

| TSAR / PAM | Low oxidation | Medium oxidation | High oxidation |
|---|---|---|---|
| Low oxidation | 0.999 | 0.904 | 0.779 |
| Medium oxidation | 0.904 | 0.999 | 0.978 |
| High oxidation | 0.773 | 0.962 | 0.999 |





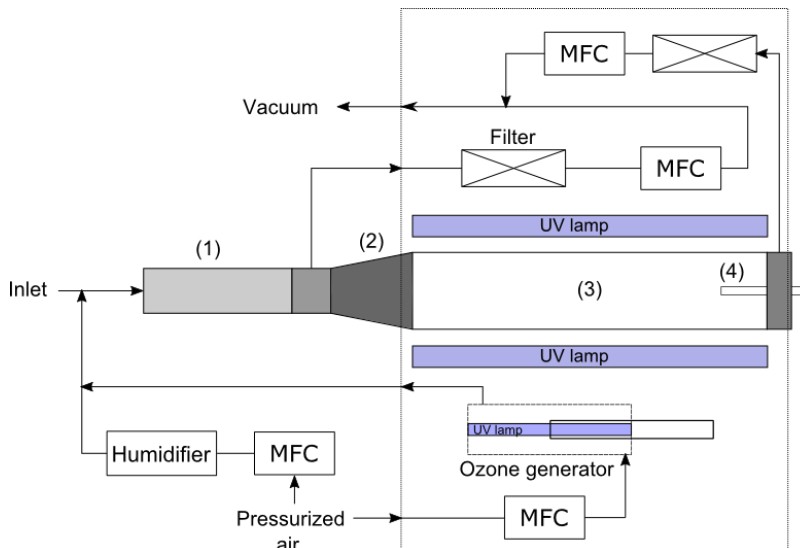

**Figure 1. TSAR layout. The residence time chamber (1), the expansion tube (2), the oxidation reactor (3) and the adjustable outlet (4).**

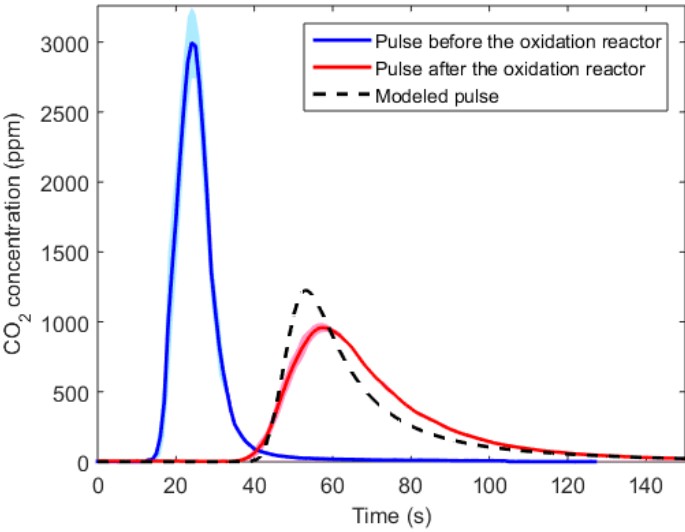

5  **Figure 2. Measured and modeled CO₂ pulses before and after the oxidation reactor. The shaded area shows the standard deviation of three pulses. The CO₂ background of 380 ppm is subtracted from the results.**



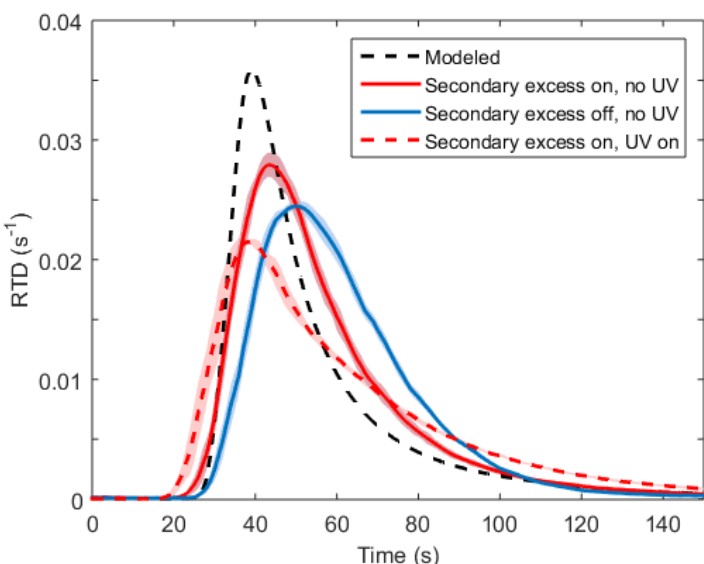

**Figure 3. Modeled and measured residence time distributions with and without secondary excess flow and UV lights. The shaded area shows the standard deviation of three pulses.**

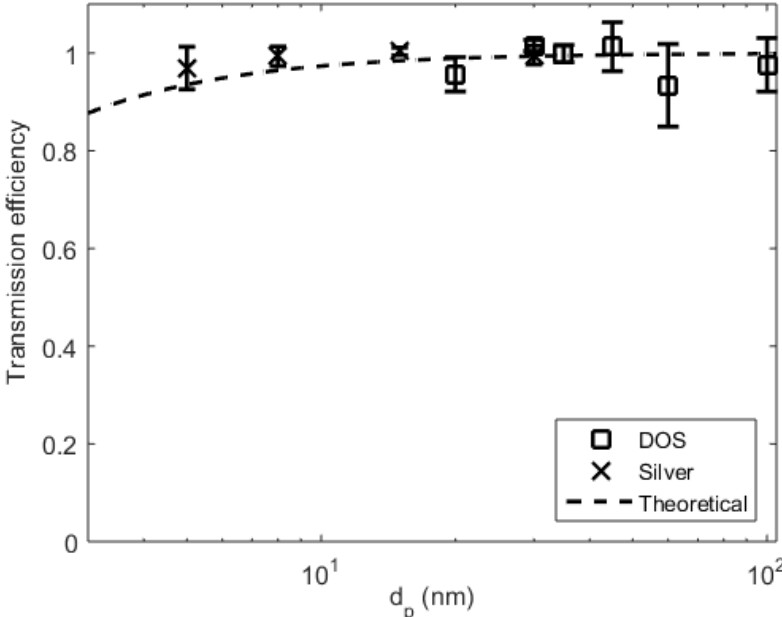

5   **Figure 4. The particle transmission efficiency in TSAR oxidation reactor. The dashed line shows the theoretical transmission efficiency when the diffusion losses are taken into account. Error bars show the standard deviation.**





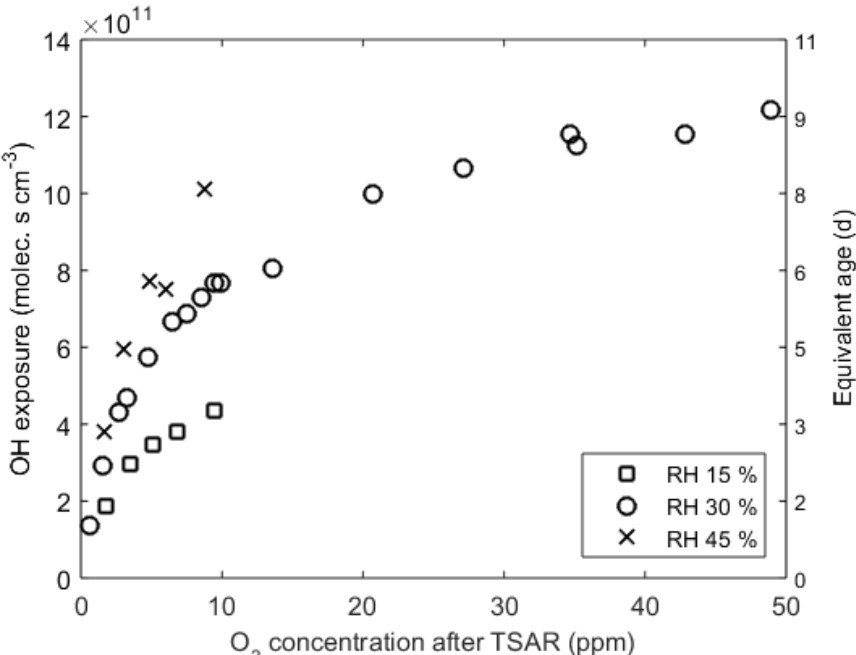

**Figure 5. The OH exposure as a function of O3 concentration after TSAR.**

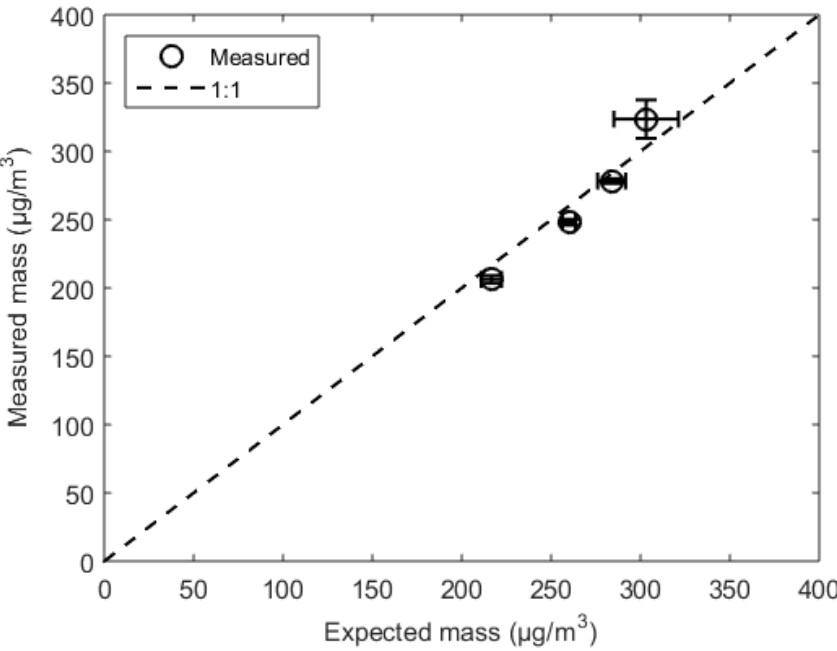

**Figure 6. Measured sulfuric acid mass as a function of expected sulfuric acid mass.**



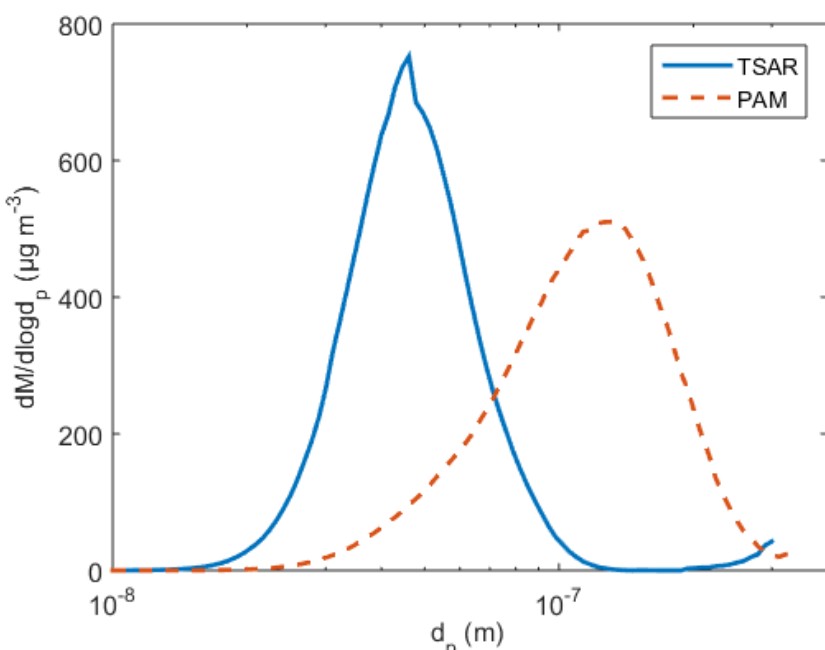

**Figure 7. Particle mass size distributions for TSAR and PAM generated toluene SOA, obtained from SMPS particle number size distribution assuming spherical particles with a density of 1.45 g cm⁻³.**

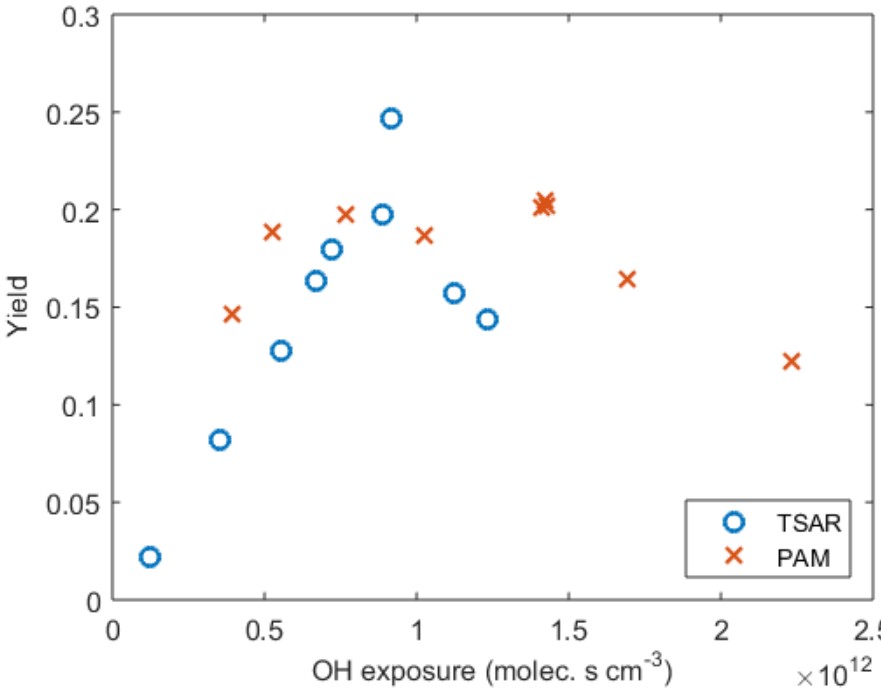

5    **Figure 8. Toluene SOA yield as a function of OH exposure for both reactors.**





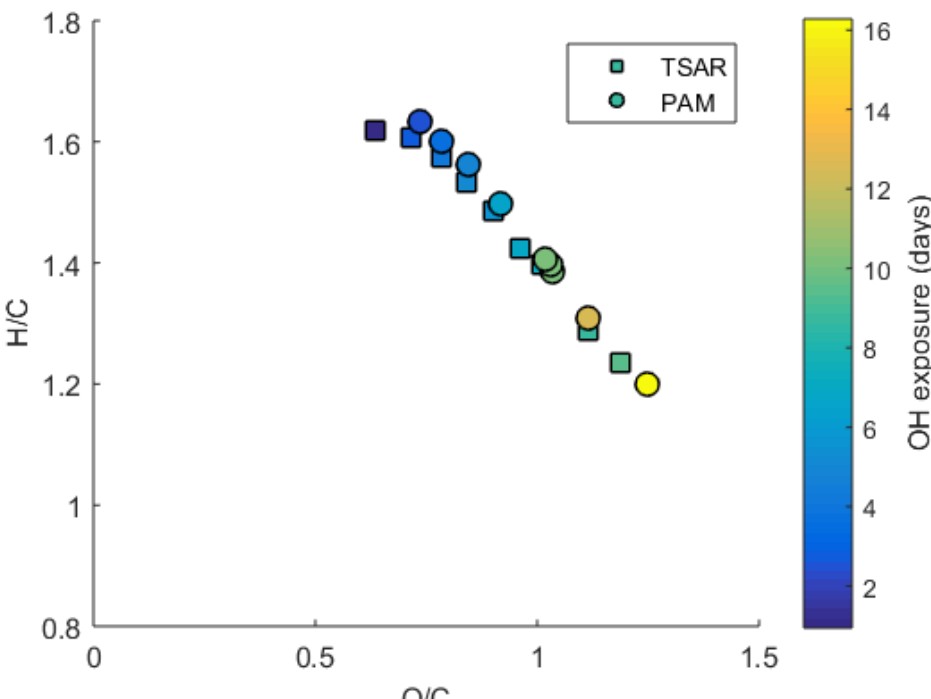

**Figure 9. The van Krevelen diagram of toluene SOA for both chambers. The color indicates the OH exposure.**

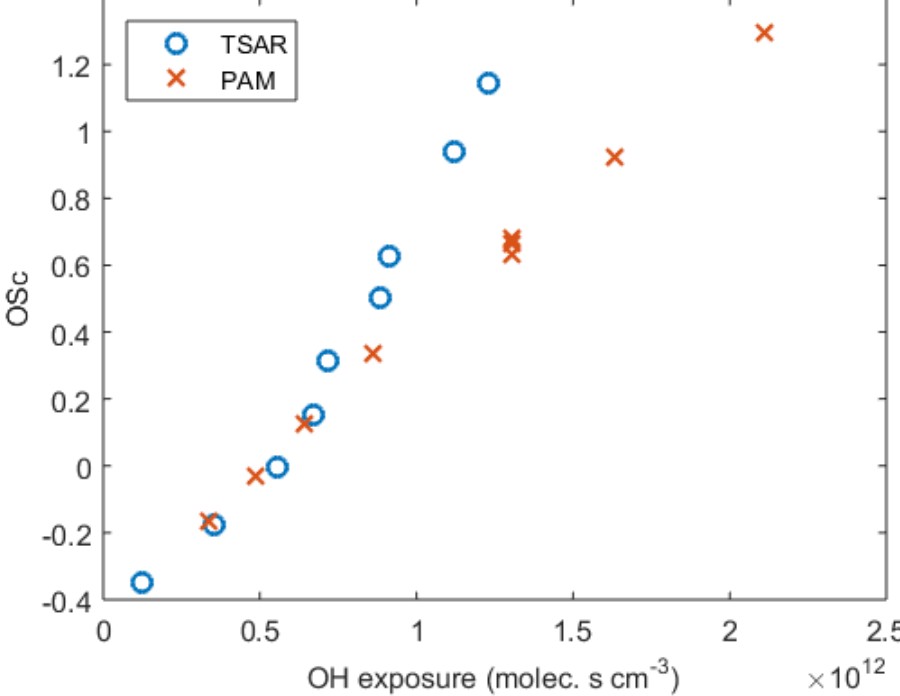

**Figure 10. The average carbon oxidaiton state ($\overline{OS_C}$) as a function of OH exposure for PAM and TSAR generated toluene SOA.**





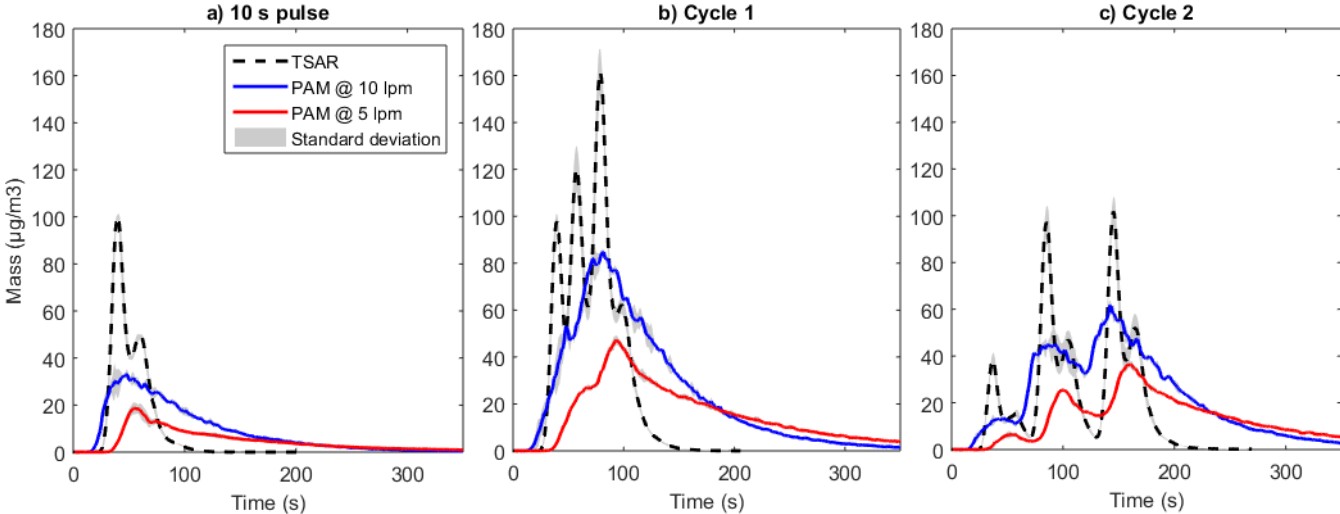

**Figure 11.** The mass produced from SOA formation of toluene pulses in TSAR at 5 slpm flow rate and in PAM at 5 and 10 slpm flow rates. The shaded area shows the standard deviation. The figures show the mass formation of a single pulse (a), cycle 1 (b) and cycle 2 (c), which both comprise of three adjacent pulses.

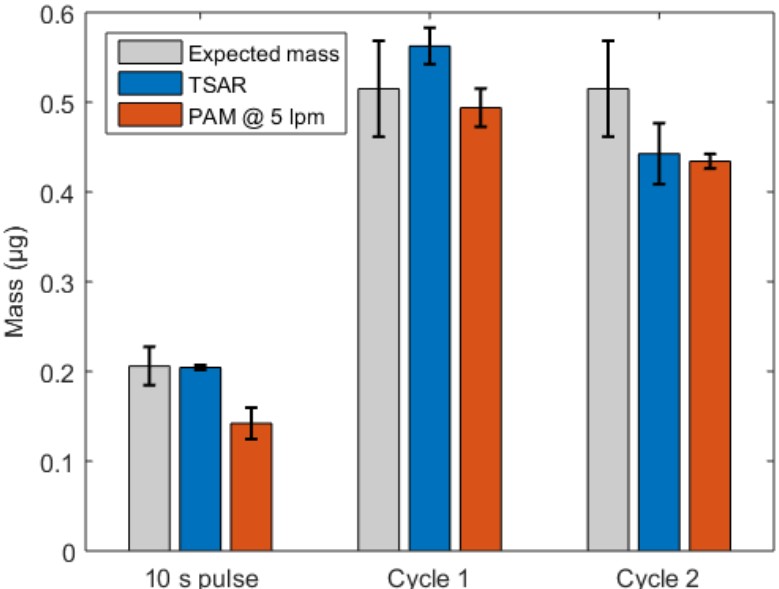

**Figure 12.** The expected and measured masses produced in pulse experiments. The expected mass is calculated from the mass of injected toluene and its SOA yield.





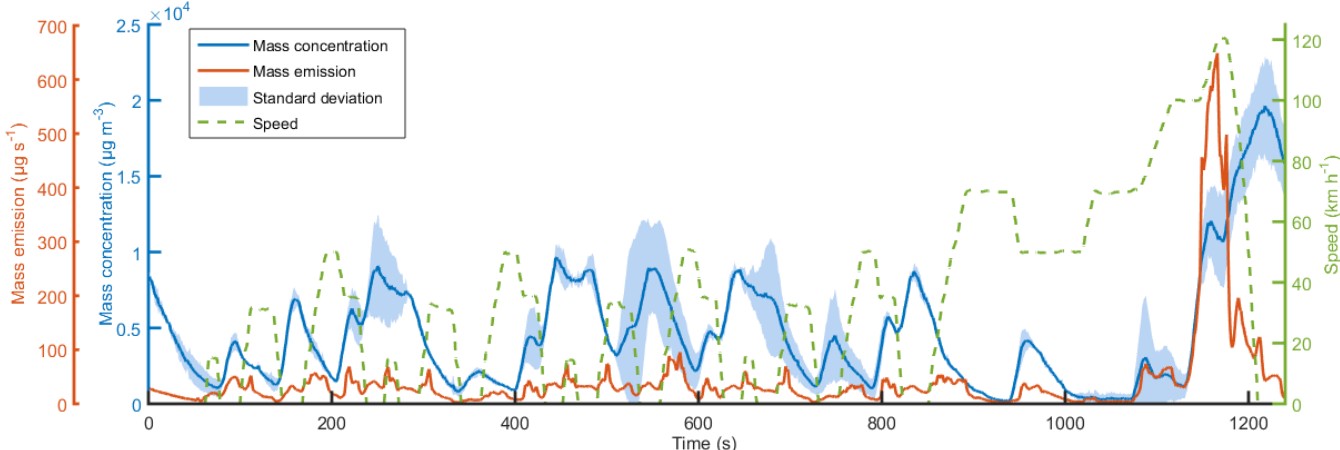

**Figure 13. Time-series of the vehicle speed, the secondary mass concentration (µg m⁻³) and the secondary mass emission factor (µg s⁻¹) during the NEDC. The emission factor is obtained by multplying the mass concentration by the total volumetric flow of the exhaust.**

