# Peer review of "The PAM OH exposure as a function of ozone concentration measured after the PAM is presented in Fig. S1. The OH exposure was determined by measuring the SO2 loss as in Sect. 2.4, and the different ozone concentrations are achieved by varying the lamp voltage."

_Atmospheric Measurement Techniques, 2016_

## Referee Comment (RC1) · Anonymous Referee #1 · 23 Dec 2016

Simonen et al. present experimental evaluation of the new Tampere Secondary Aerosol Reactor (TSAR). Performance metrics of the TSAR are evaluated against the Potential Aerosol Mass (PAM) oxidation flow reactor: residence time distributions, particle losses, OH exposure, sulfuric acid and toluene SOA yields, and O/C and H/C of toluene SOA. The authors additionally demonstrate the application of the TSAR to resolve changes in the SOA formation potential of car exhaust during dynamometer measurements of a vehicle subjected to a transient driving cycle. The TSAR has potential applications for secondary aerosol formation potential studies of emissions sources. Overall, I was impressed. I would support publication in Atmospheric Measurement Techniques after consideration of my comments below.

1. The authors present evidence that the shorter residence time, higher particle transmission efficiency, and narrower residence time distributions of the TSAR relative to the PAM reactor provide several advantages, especially for transient drive cycle measurements and similar applications. However, in my opinion the short residence time (37 sec) also presents important limitations that are not adequately discussed in the current manuscript. In the absence of significant seed particle surface area, such a short residence time likely limits the condensation of oxidized organic species, to the point that SOA yields that are reported from the TSAR in principle represent lower limits compared SOA yields reported from oxidation flow reactors with longer residence times and/or smog chambers.

   For example, the accommodation coefficient ($\alpha$) has been shown to be approximately 0.1 for some oxidized organic vapors that condense to form SOA (e.g. Saleh et al.., 2013). To the extent that $\alpha = 0.1$ is representative of other SOA types, this suggests that the relevant timescales for condensation of oxidized organic vapors ranges from minutes to hours (e.g. Lambe et al., 2015; Seinfeld and Pandis, 1998). This is likely less of an issue in the TSAR for species such as sulfuric acid ($\alpha \sim 1$), which may be why the authors obtain mass closure with sulfuric acid. The following results presented by Simonen et al. further support the hypothesis that condensation of oxidized organic vapors is limited by the short residence time:

   a. Particle size distributions of toluene SOA generated in TSAR and PAM reactor at comparable OH exposure (Fig. 7) show that the mode diameter of SOA particles generated in the TSAR is much smaller than in the PAM reactor (40 nm versus 100 nm).
   b. The maximum toluene SOA yield value obtained in the PAM reactor would increase from ~0.20 to ~0.33 (Fig. 8) if size-dependent particle loss corrections are applied (from statement on Page 14, Line 17, I assume the yields are not currently corrected for losses). This correction is obtained by applying a measured transmission efficiency ~0.6 for 100 nm dioctyl sebacate particles through the PAM reactor (Fig. 2; Lambe et al., 2011). The yield values obtained at other OH exposures would presumably also increase significantly, indicating that toluene SOA yields in the PAM reactor were ~40% or higher than toluene SOA yields measured in the TSAR. Importantly, the mean residence time of the PAM reactor operated in these studies was also significantly higher (~160 sec, as calculated from internal volume of 13.3 L and flow rate 5 LPM).

   In my opinion, the authors should add a paragraph near the beginning of the paper that states this up front, and discusses the potential implications of a very short residence time – perhaps at the end of Section 2.1, and elsewhere in the manuscript as appropriate. I think it is also worth

attempting to constrain the magnitude of residence-time-limited condensational growth, so that reported TSAR SOA yields obtained with a 37 sec residence time can be corrected (with some uncertainty) to SOA yields that would be obtained with longer condensation times. For example, the authors might consider implementing the "LVOC fate correction" model described in Palm et al. (2016) or something similar to estimate the magnitude of condensable vapors that exit the TSAR before condensing elsewhere.

2. **Page 3, Section 2.1**: It may be useful to briefly state why the authors choose to operate the TSAR as an "OFR254" instead of an "OFR185". I assume that "OFR254" is more practical/safer because the lamps are external to the reactor, and the 185 nm light is not transmitted through the quartz walls of the reactor? Are there additional reasons? Is the UV intensity inside the TSAR varied at all, or is the OH exposure varied by changing the $O_3$ and/or $H_2O$ concentrations?

3. **Page 5, Lines 9-10**: The authors state: "the only significant loss of SO2 in the oxidation reactor is due to the reaction with OH radicals." Was this assumed, or verified experimentally? Was the transmission efficiency of $SO_2$ measured in the absence of OH? Please clarify.

4. **Page 6, Line 4**: The authors state: "injecting pressurized air, SO2, humidified air and ozone into the TSAR". This is a confusing – isn't the pressurized air also humidified? Figure 1 suggests that some fraction of the carrier gas is sent through a humidifier, another fraction is sent through the ozone generator, and both are mixed prior to entering the residence time chamber.

5. **Page 6, Line 27**: Consider deleting "which represent atmospheric oxidation" -- in my opinion there are open questions as to how representative all laboratory simulations of atmospheric oxidation chemistry are (flow reactors and smog chambers).

6. **Page 7, Line 9**: Please explain why the PAM reactor was run in "OFR185" mode and not also "OFR254" mode to more closely represent the TSAR operating conditions.

7. **Page 8, Line 2**: Consider replacing "theoretical residence time" with "calculated mean residence time" or "mean plug-flow residence time", or something similar.

8. **Page 10, Lines 5-9**: The added convection caused by UV lamp-induced temperature gradients can be modeled by Taylor dispersion (e.g. Lambe et al., 2011; Huang and Coggon et al., 2016). I suggest applying a Taylor dispersion model to see if this more closely represents the measured residence time distributions that are presented in Figure 3.

9. **Page 11, Line 9**: I did not notice where the output of the UV lamps is discussed. Please report the UV actinic flux that is currently used in the TSAR. This is needed to get at least a qualitative sense of the relative importance of photolysis versus OH oxidation pathways for selected species (e.g. Peng et al., 2016). It should be possible to constrain the UV actinic flux from the measured OH exposures and input ozone and water vapor mixing ratios using the model described in Li et al. (2015) or something similar.

10. **Page 11, Section 3.4**: The authors might consider combining some of the text from Section 2.5 with Section 3.4, i.e. Page 5 Lines 25-29, Page 6 Lines 1-2 and Lines 10-20 to consolidate the discussion of sulfuric acid yield measurements.

11. **Page 11, Section 3.5**: The authors might consider combining some of the text from Section 2.6 (Page 6, Lines 22-30) with Section 3.5.

12. **Page 12, Lines 5-7**: This argument is speculative and not supported by the measurements. If it were true that "the PAM always outputs SOA with both high $OH_{exp}$ and low $OH_{exp}$, [thereby reducing] the sensitivity to $OH_{exp}$", I would argue that the range of O/C and H/C of PAM-generated toluene SOA should also be narrower than in the TSAR. Figures 9 and 10 show that this is not the case – the range of O/C and H/C is similar in both reactors. I would consider deleting this statement. A more plausible argument, in my opinion, is that the residence-time-limited condensation of oxidized vapors in the TSAR (mean residence time = 37 sec) compared to the PAM reactor (mean residence time ~ 160 sec) results in higher sensitivity to changes in yields of condensable species at the lowest and highest OH exposures that were studied.

13. **Page 12, Line 29 – Page 13, Line 5:** Here, it may be useful to know the UV actinic flux in both reactors (see Comment #9). For example, at $OH_{exp}$ ~ $1.3*10^{12}$ molec $cm^{-3}$ s, if "F254/OHexp" (as defined by Peng et al., 2016) is significantly different in the TSAR and in the PAM reactor, one might hypothesize that direct UV photolysis of the SOA is more important in one system than the other, and perhaps this is correlated with the observation that OSc ~ 1.1 in the TSAR and OSc ~ 0.6 in the PAM reactor (Fig. 10). It is hard to tell without knowledge of the actinic fluxes in both reactors. It may also be useful to add error bars to represent the approximate uncertainties in OH exposure in both systems.

14. **Page 9, Section 2.7, Lines 4-6 and Pages 15-16, Section 3.6.2:** In Section 2.7, the authors state: "the amount of secondary aerosol mass produced in the TSAR was determined by subtracting the primary mass from the mass measured when using the TSAR." The absolute secondary-to-primary aerosol mass enhancement is actually never discussed. I think it would be useful to show time series of the primary mass concentration and the secondary-to-primary emission ratio in Figure 13 (perhaps as additional subpanels), discuss in Section 3.6.2, and briefly compare with results from relevant literature studies such as Platt et al. (2013), Tkacik et al. (2014) and Karjalainen et al. (2016).

**References**

R. Saleh, N. M. Donahue, and Robinson, A. L.: Time scales for gas-particle partitioning equilibration of secondary organic aerosol formed from $\alpha$-pinene ozonolysis, Environ. Sci. Technol., 47, 5588–5594, 2013.

B.B. Palm, P. Campuzano-Jost, A.M. Ortega, D.A. Day, L. Kaser, W. Jud, T. Karl, A. Hansel, J.F. Hunter, E.S. Cross, J.H. Kroll, A. Turnipseed, Z. Peng, W.H. Brune, and J.L. Jimenez. In situ secondary organic aerosol formation from ambient pine forest air using an oxidation flow reactor. *Atmospheric Chemistry and*

*Physics*, Atmos. Chem. Phys., 16, 2943-2970, doi:10.5194/acp-16-2943-2016, 2016. Source code available for download at https://sites.google.com/site/pamwiki/hardware/estimation-equations.

Y. Huang, M. M. Coggon , R. Zhao, H. Lignell, M. U. Bauer, R. C. Flagan, and J. H. Seinfeld. The Caltech Photooxidation Flow Tube Reactor − I : Design and Fluid Dynamics. Atmos. Meas. Tech. Discuss., doi:10.5194/amt-2016-282, 2016.

---

## Referee Comment (RC2) · Anonymous Referee #2 · 22 Jan 2017

General comments:

The manuscript by Simonen et al., describes a new oxidation flow reactor designed to achieve shorter residence time relative to the potential aerosol mass (PAM) reactor. The aim of the new reactor is to provide a method of capturing SOA formation during studies of rapidly changing emission sources (e.g. combustion emissions). Although the subject of the manuscript is appropriate for publication in Atmospheric Measurement Techniques, the manuscript in its current state reads more like a technical report at certain parts and lacks sufficient discussion in many places. Most importantly, the manuscript does not contain sufficient emphasis on the atmospheric applicability or relevance in several places. In general, the use of such methodology for studying SOA

formation has some benefit if results are not over-interpreted, and used for comparative purposes or during screening experiments to quantify the "potential" SOA formation of a given source or precursor. This is because the methodology is fundamentally limited in terms of its ability to reproduce tropospheric conditions due to unrealistic partitioning behaviour at elevated supersaturation of oxidation products, high OH exposure over a very short period of time and complex OH vs non-OH chemistry resulting from high photon flux at non-tropospheric wavelengths, especially for mixtures of high OH reactivity. Consequently, results of such experiments must not be over-interpreted or used in absolute term. It is recommended that the authors address the general and specific comments adequately before the manuscript is considered for publication.

Specific comments:

1) In the abstract, the statement about long and short residence times of different reactors should be qualified by adding typical times to provide the reader with an idea about the main difference between existing techniques and the new TSAR reactor. The abstract should also contain more details about the main modification of the reactor design, which enables the operation at higher time resolution compared to the PAM reactor (e.g. volume, flow rate).

2) The introduction should include more critical evaluation of the limitations of existing reactors and chambers with regards to the ability to measure transients and provide typical examples of the residence times of the various reactors mentioned in order to make a case for the need of the new reactor being described in this manuscript.

3) Page 4, line 1-2: The O3 needed for this reaction chain is mixed with the sample prior to the residence time chamber. This has an implication on experiments where some of the VOCs react with ozone (e.g. alkenes and biogenic compounds). This should be stated and clarified with discussion of how it would or wouldn't be possible to separate such an effect from that of OH chemistry especially for mixtures of complex composition (i.e. real emissions). The authors touch on this effect later in the manuscript when

comparing the oxidation state of SOA produced by the TSAR and PAM reactors (page 12, line 26-27), which emphasise the importance of characterising this aspect of the reactor.

4) Page 6, line 19-20: Although the authors have shown that the assumption that sulfuric acid losses to the reactor wall are negligible, it is not clear whether this is meant to suggest that such assumption would also hold for oxidation products of organic compounds, which have a wider range of volatility distribution. This should be discussed in the manuscript as it represents a limitation on the ability to quantify SOA yield. On a related note, the sulfuric acid yield section (3.4) is too short to stand alone as it is. The discussion need to be expanded to address this comment.

5) Page 7, line 9: It is not clear why the authors did not use the PAM reactor in the OFR254 mode given intended purpose of comparing the results with TSAR which uses a 254nm light source. This should be explained and justified. In addition, the manuscript provides no discussion of the potential effect of the different light sources on the non-OH chemistry in the reactors. The authors should include such discussion in the manuscript in the context of the work published by Peng et al., (2016) quantifying the extent of OH vs. non-OH chemistry according to the conditions applied in the reactor. It is important to understand the role of water mixing ratio, photon flux and external OH reactivity in the experiments on non-OH chemistry (photolysis) in order to establish the atmospheric relevance of the experiments.

6) Page 11, line 21: What is the background mass of the TSAR? How variable is it depending on OH and humidity conditions? Is this characterised and corrected for on a regular basis? More discussion of this should be included.

7) Page 11, line 25-28: More discussion is needed for the apparent link between shorter residence time and the smaller size distribution produced in the TSAR experiment. The potential implication of such phenomena on the produced SOA particles and their properties needs to also be discussed.

8) Page 12, line 1-2: How representative is the off-line OH exposure calibration of the actual reported OH exposure in a more complex VOC mixture such that found in the Toluene SOA experiment or other VOC mixtures with different OH reactivates? This is likely to be a source of significant uncertainty in the determination of OH exposure and it is not discussed adequately in the manuscript.

9) Page 12, line 9-15: The discussion of the different toluene SOA yields among the Kang et al., Ng et al., and Hilderbrandt et al., is very brief and over-simplified. There are so many factors affecting the different studies that could potentially contribute to the reported SOA yields and differences cannot be explained only be the presence or absence of seed particles.

10) Page 12, line 18: The Aiken et al., (2008) analysis method for HR-AMS data has been updated by Canagaratna et al., (2015), with the new method having a direct effect on the reported O:C values. The authors should either justify the reason why the opted to use the Aiken calibration or update the results using the Canagaratna method.

Editorial comments:

Page 4, line 11: section should be 2.2 not 2.1 (same correction should be applied to all subsequent subsections in this part of the manuscript). Page 10, line 14-16: the temperature and relative humidity should be reported as average with an associated standard deviation instead of the current mixing up of average with range.

References:

Peng et al., (2016), Non-OH chemistry in oxidation flow reactors for the study of atmospheric chemistry systematically examined by modeling. Atmos. Chem. Phys., 16(7), 4283-4305. doi:10.5194/acp-16-4283-2016

---

## Author Comment (AC1)

**Response to referees**

We thank both referees for constructive comments and suggestions.

Especially the comments on the possible problems arising from the short residence time are very relevant for this study. We believe the improvements we made concerning this topic and other comments significantly add to the quality of the paper.

Since the changes apply to almost every section of the manuscript, we formulate the response as follows:

- The referee comments are shown as blue text, and the corresponding responses as black text.
- The exact changes made to the manuscript are not included in the responses, but instead we attach the new manuscript and supplementary material at the end of the document. The changes are shown as red text.

**Response to Anonymous Referee #1**

Simonen et al. present experimental evaluation of the new Tampere Secondary Aerosol Reactor (TSAR). Performance metrics of the TSAR are evaluated against the Potential Aerosol Mass (PAM) oxidation flow reactor: residence time distributions, particle losses, OH exposure, sulfuric acid and toluene SOA yields, and O/C and H/C of toluene SOA. The authors additionally demonstrate the application of the TSAR to resolve changes in the SOA formation potential of car exhaust during dynamometer measurements of a vehicle subjected to a transient driving cycle. The TSAR has potential applications for secondary aerosol formation potential studies of emissions sources. Overall, I was impressed. I would support publication in Atmospheric Measurement Techniques after consideration of my comments below.

1. The authors present evidence that the shorter residence time, higher particle transmission efficiency, and narrower residence time distributions of the TSAR relative to the PAM reactor provide several advantages, especially for transient drive cycle measurements and similar applications. However, in my opinion the short residence time (37 sec) also presents important limitations that are not adequately discussed in the current manuscript. In the absence of significant seed particle surface area, such a short residence time likely limits the condensation of oxidized organic species, to the point that SOA yields that are reported from the TSAR in principle represent lower limits compared SOA yields reported from oxidation flow reactors with longer residence times and/or smog chambers.

For example, the accommodation coefficient ( $\alpha$ ) has been shown to be approximately 0.1 for some oxidized organic vapors that condense to form SOA (e.g. Saleh et al., 2013). To the extent that  $\alpha = 0.1$  is representative of other SOA types, this suggests that the relevant timescales for condensation of oxidized organic vapors ranges from minutes to hours (e.g. Lambe et al., 2015; Seinfeld and Pandis, 1998). This is likely less of an issue in the TSAR for species such as sulfuric acid ( $\alpha \sim 1$ ), which may be why the authors obtain mass closure with sulfuric acid. The following results presented by Simonen et al. further support the hypothesis that condensation of oxidized organic vapors is limited by the short residence time:

a. Particle size distributions of toluene SOA generated in TSAR and PAM reactor at comparable OH exposure (Fig. 7) show that the mode diameter of SOA particles generated in the TSAR is much smaller than in the PAM reactor (40 nm versus 100

nm).

b. The maximum toluene SOA yield value obtained in the PAM reactor would increase from ~0.20 to ~0.33 (Fig. 8) if size-dependent particle loss corrections are applied (from statement on Page 14, Line 17, I assume the yields are not currently corrected for losses). This correction is obtained by applying a measured transmission efficiency ~0.6 for 100 nm dioctyl sebacate particles through the PAM reactor (Fig. 2; Lambe et al., 2011). The yield values obtained at other OH exposures would presumably also increase significantly, indicating that toluene SOA yields in the PAM reactor were ~40% or higher than toluene SOA yields measured in the TSAR. Importantly, the mean residence time of the PAM reactor operated in these studies was also significantly higher (~160 sec, as calculated from internal volume of 13.3 L and flow rate 5 LPM).

In my opinion, the authors should add a paragraph near the beginning of the paper that states this up front, and discusses the potential implications of a very short residence time – perhaps at the end of Section 2.1, and elsewhere in the manuscript as appropriate. I think it is also worth attempting to constrain the magnitude of residence-time-limited condensational growth, so that reported TSAR SOA yields obtained with a 37 sec residence time can be corrected (with some uncertainty) to SOA yields that would be obtained with longer condensation times. For example, the authors might consider implementing the "LVOC fate correction" model described in Palm et al. (2016) or something similar to estimate the magnitude of condensable vapors that exit the TSAR before condensing elsewhere.

The limited time for the condensing of oxidized species is now discussed in Sect. 2.5 and the effects are modeled in Sect. 3.4. The effects are also studied in the sections where it is relevant (3.5 and 3.6). Indeed, the LVOC losses are higher in TSAR because of the shorter residence time, especially when the condensational sink is low.

The condensational sink in toluene experiments was high, so that the modeled losses were generally 2 % - 35 %, depending on the value of the accommodation coefficient. Even with the lowest accommodation coefficient (0.1), there was no drastic difference between the losses in PAM and TSAR: at  $2 \times 10^{11} < OH_{exp} < 9 \times 10^{11}$  molec. cm-3 s, the losses in TSAR were ~10 % and in PAM ~7%. At higher and lower OHexp the TSAR losses increase faster than the PAM losses.

The yields were not corrected for the particle losses since the correction for nucleated particles which grow in the PAM is not straightforward. However, we added a paragraph considering the PAM losses, and found that when corrected with the transmission efficiency measured for this particular PAM reactor, the yield is ~19 % higher with the correction (Karjalainen et al., 2016).

2. Page 3, Section 2.1: It may be useful to briefly state why the authors choose to operate the TSAR as an "OFR254" instead of an "OFR185". I assume that "OFR254" is more practical/safer because the lamps are external to the reactor, and the 185 nm light is not transmitted through the quartz walls of the reactor? Are there additional reasons? Is the UV intensity inside the TSAR varied at all, or is the OH exposure varied by changing the O3 and/or H2O concentrations?

You are right, the OFR254 suits better the TSAR because of the limited transmittance at 185 nm wavelength of the quartz glass. Even if different glass was used, the use of OFR185 mode would require a special ventilation for the TSAR casing to avoid the ozone formation in the room air. This is now clarified in Sect. 2.1. The UV intensity is not varied, and this is clarified in Sect. 2.1 also by describing the lamps as "constant power ozone free low pressure mercury lamps". However, the

intensity control will be implemented in future to avoid high photon exposure to OH exposure ratio which can cause unwanted photolysis.

3. Page 5, Lines 9-10: The authors state: "the only significant loss of SO2 in the oxidation reactor is due to the reaction with OH radicals." Was this assumed, or verified experimentally? Was the transmission efficiency of SO2 measured in the absence of OH? Please clarify.

The  $SO_2$  transmission efficiency was not measured. However, the possible wall loss does not affect the calculation because the  $SO_2$  concentration is always measured after TSAR, with or without UV lights. Thus, the wall loss term in the differential equation cancels out (assuming that the wall loss does not depend on UV lights). This is now clarified in the text.

4. Page 6, Line 4: The authors state: "injecting pressurized air, SO2, humidified air and ozone into the TSAR". This is a confusing – isn't the pressurized air also humidified? Figure 1 suggests that some fraction of the carrier gas is sent through a humidifier, another fraction is sent through the ozone generator, and both are mixed prior to entering the residence time chamber.

Yes, a fraction of the carrier gas is sent through a humidifier and the SO2 is diluted with pressurized air. The sentence was rewritten as follows: "injecting humidified air, ozone and SO2 diluted with pressurized air".

5. Page 6, Line 27: Consider deleting "which represent atmospheric oxidation" -- in my opinion there are open questions as to how representative all laboratory simulations of atmospheric oxidation chemistry are (flow reactors and smog chambers).

This statement is now removed and discussion about the atmospheric representativeness is added to the Introduction, as recommended by Referee #2.

6. Page 7, Line 9: Please explain why the PAM reactor was run in "OFR185" mode and not also "OFR254" mode to more closely represent the TSAR operating conditions.

We wanted to compare two different systems that produce the same OH exposure to see if TSAR could be used in same applications as PAM, which is usually operated in OFR185 mode at least in engine exhaust measurements (Karjalainen et al., 2016; Timonen et al., 2016; Tkacik et al., 2014). This motivation is now added to the manuscript.

7. Page 8, Line 2: Consider replacing "theoretical residence time" with "calculated mean residence time" or "mean plug-flow residence time", or something similar.

"Theoretical residence time" is now replaced with "mean plug-flow residence time". There was also a mistake in this section: PAM is not twice as big as TSAR, but the PAM volume is approximately four times the volume of TSAR oxidation reactor. This is also corrected in the manuscript.

8. Page 10, Lines 5-9: The added convection caused by UV lamp-induced temperature gradients can be modeled by Taylor dispersion (e.g. Lambe et al., 2011; Huang and Coggon et al., 2016). I suggest applying a Taylor dispersion model to see if this more closely represents the measured residence time distributions that are presented in Figure 3.

According to Huang and Coggon et al. (2016), the Taylor dispersion approximation applies when

$$\tau_{c,cyld} \gg \frac{\tau_{c,D_i}}{2.82^2}$$

where  $\tau_{c_i D_i} = \frac{R^2}{D_i}$  and  $\tau_{c_i cyld} = \frac{L_{cyld}}{U_{avg}}$ . In TSAR,  $\tau_{c_i cyld} = 39 s$  and  $\frac{\tau_{c_i D_i}}{3.83^2} = 14 s$  if  $D_i = 1 \times 10^{-5} m^2 s^{-1}$ . Thus, we think the use of Taylor dispersion approximation is not justified here.

10. Page 11, Line 9: I did not notice where the output of the UV lamps is discussed. Please report the UV actinic flux that is currently used in the TSAR. This is needed to get at least a qualitative sense of the relative importance of photolysis versus OH oxidation pathways for selected species (e.g. Peng et al., 2016). It should be possible to constrain the UV actinic flux from the measured OH exposures and input ozone and water vapor mixing ratios using the model described in Li et al. (2015) or something similar.

Thank you for this suggestion. We used a photochemical model available in PAM users manual (https://sites.google.com/site/pamusersmanual/7-pam-photochemistry-model) and reproduced the measurement results in Sect. 3.3 using the photon flux as one of the fitting parameters. The 254 nm photon flux obtained by fitting was approximately **1.92** × **10**15 photons cm-2 s-1.

Based on this photon flux, we also estimate the importance of photolysis in Sect. 3.4.1 as recommended by Referee #2. In addition, the model allows us to estimate better the OH exposure in toluene experiments, and we think this improves Sect. 3.5 significantly (see response to Referee #2, comment 8).

11. Page 11, Section 3.4: The authors might consider combining some of the text from Section 2.5 with Section 3.4, i.e. Page 5 Lines 25-29, Page 6 Lines 1-2 and Lines 10-20 to consolidate the discussion of sulfuric acid yield measurements.

As Referee #2 also pointed out, this section was short and stand-alone. The sulfuric acid measurements are now included in Sect. 3.4.2 to test the LVOC loss model validity.

12. Page 11, Section 3.5: The authors might consider combining some of the text from Section 2.6 (Page 6, Lines 22-30) with Section 3.5.

The beginning of Sect. 3.5 was quite abrupt, and this is corrected now by adding the following sentence at the beginning of Sect. 3.5: "The SOA formation studies were conducted as described in Sect. 2.6."

13. Page 12, Lines 5-7: This argument is speculative and not supported by the measurements. If it were true that "the PAM always outputs SOA with both high OHexp and Iow OHexp, [thereby reducing] the sensitivity to OHexp", I would argue that the range of O/C and H/C of PAM-generated toluene SOA should also be narrower than in the TSAR. Figures 9 and 10 show that this is not the case – the range of O/C and H/C is similar in both reactors. I would consider deleting this statement. A more plausible argument, in my opinion, is that the residence-time-limited condensation of oxidized vapors in the TSAR (mean residence time = 37 sec) compared to the PAM reactor (mean residence time ~ 160 sec) results in higher sensitivity to changes in yields of condensable species at the lowest and highest OH exposures that were studied.

We agree that this argument was not supported by the O/C and H/C measurements. After estimating the OH exposure by using the photochemical model, the difference in TSAR and PAM yields is not

anymore so big. There is a difference now only at low OH exposures, but the uncertainty of PAM yields in those cases is very high. Thus, the speculation of PAM OH exposure distribution was removed.

14. Page 12, Line 29 – Page 13, Line 5: Here, it may be useful to know the UV actinic flux in both reactors (see Comment #9). For example, at OHexp ~ 1.3\*1012 molec cm-3 s, if "F254/OHexp" (as defined by Peng et al., 2016) is significantly different in the TSAR and in the PAM reactor, one might hypothesize that direct UV photolysis of the SOA is more important in one system than the other, and perhaps this is correlated with the observation that OSc ~ 1.1 in the TSAR and OSc ~ 0.6 in the PAM reactor (Fig. 10). It is hard to tell without knowledge of the actinic fluxes in both reactors. It may also be useful to add error bars to represent the approximate uncertainties in OH exposure in both systems.

After estimating the OH exposure by using the photochemical model, the difference in TSAR and PAM OSc is not as high as before, even though the trend seems to be different between these reactors. However, after applying the error bars arising from the uncertainty of the model, we cannot say whether the difference in trends is real. The error bars for OH exposure are now added to this figure and also to the yield and O/C-H/C figure (Figs. 10,11 and 12 in the new manuscript).

15. Page 9, Section 2.7, Lines 4-6 and Pages 15-16, Section 3.6.2: In Section 2.7, the authors state:

"the amount of secondary aerosol mass produced in the TSAR was determined by subtracting the primary mass from the mass measured when using the TSAR." The absolute secondaryto-primary aerosol mass enhancement is actually never discussed. I think it would be useful to show time series of the primary mass concentration and the secondary-to-primary emission ratio in Figure 13 (perhaps as additional subpanels), discuss in Section 3.6.2, and briefly compare with results from relevant literature studies such as Platt et al. (2013), Tkacik et al. (2014) and Karjalainen et al. (2016).

**Response to Anonymous Referee #2**

**General comments:**

The manuscript by Simonen et al., describes a new oxidation flow reactor designed to achieve shorter residence time relative to the potential aerosol mass (PAM) reactor. The aim of the new reactor is to provide a method of capturing SOA formation during studies of rapidly changing emission sources (e.g. combustion emissions). Although the subject of the manuscript is appropriate for publication in Atmospheric Measurement Techniques, the manuscript in its current state reads more like a technical report at certain parts and lacks sufficient discussion in many places. Most importantly, the manuscript does not contain sufficient emphasis on the atmospheric applicability or relevance in several places. In general, the use of such methodology for studying SOA formation has some benefit if results are not over-interpreted, and used for comparative purposes or during screening experiments to quantify the "potential" SOA formation of a given source or precursor. This is because the methodology is fundamentally limited in terms of its ability to reproduce tropospheric conditions due to unrealistic partitioning behaviour at elevated supersaturation of oxidation products, high OH exposure over a very short period of time and complex OH vs non-OH chemistry resulting from high photon flux at non-tropospheric wavelengths, especially for mixtures of high OH reactivity. Consequently, results of such experiments must not be over-interpreted or used in absolute term. It is recommended that the authors address the general and specific comments adequately before the manuscript is considered for publication.

The discussion in the manuscript is now extended by considering the atmospheric applicability in the Introduction and by considering the effects caused by the short residence time and photolysis in new sections (*2.5 Estimating vapor losses and photolysis* and *3.4 Vapor losses and photolysis in TSAR*). The results from these sections are also applied to Sect. 3.5 and 3.6.

We agree that the flow reactors do not reproduce tropospheric conditions and that their best application is comparative measurements, such as Timonen et al. (2016) where the effect of fuel ethanol content on the exhaust SOA potential was studied using a PAM reactor. However, there is no method that could fully reproduce the conditions in the atmosphere, but still estimations are needed e.g. for emission inventories. Thus, we think also absolute values should be published in similar way as when using smog chambers (e.g. Gordon et al., 2014; Nordin et al., 2013; Platt et al., 2013). This allows comparison of the values also between studies (when the oxidation conditions are comparable). In any case, an adequate error analysis and analysis on atmospheric applicability must be done. This discussion is now added to Introduction and Sect. 2.5 and 3.4.

**Specific comments:**

1) In the abstract, the statement about long and short residence times of different reactors should be qualified by adding typical times to provide the reader with an idea about the main

difference between existing techniques and the new TSAR reactor. The abstract should also contain more details about the main modification of the reactor design, which enables the operation at higher time resolution compared to the PAM reactor (e.g. volume, flow rate).

The typical times are now added to the abstract, and also the main difference that enables the higher time resolution compared to the PAM (smaller radius and consequently smaller volume).

2) The introduction should include more critical evaluation of the limitations of existing reactors and chambers with regards to the ability to measure transients and provide typical examples of the residence times of the various reactors mentioned in order to make a case for the need of the new reactor being described in this manuscript.

The motivation for developing TSAR is now clarified in the introduction. We also added discussion about micro smog chamber (MSC, Keller and Burtscher, 2012) which has even shorter residence time than TSAR. In the introduction we discuss why a compromise between PAM and MSC is needed.

3) Page 4, line 1-2: The O3 needed for this reaction chain is mixed with the sample prior to the residence time chamber. This has an implication on experiments where some of the VOCs react with ozone (e.g. alkenes and biogenic compounds). This should be stated and clarified with discussion of how it would or wouldn't be possible to separate such an effect from that of OH chemistry especially for mixtures of complex composition (i.e. real emissions). The authors touch on this effect later in the manuscript when comparing the oxidation state of SOA produced by the TSAR and PAM reactors (page 12, line 26-27), which emphasise the importance of characterising this aspect of the reactor.

This effect was already mentioned very shortly at the end of Sect. 3.5.1 (p.13, line 20-23 in original manuscript). The discussion is now extended at the end of Sect. 3.5.1 (p. 21, line 10-17). The main application of TSAR is to sample vehicular emissions, and thus the ozone reactive biogenic VOCs are not of concern here. SOA precursors in vehicle emissions are dominated by (non-alkene) compounds that are unreactive toward ozone (Gentner et al., 2012; Tkacik et al., 2014).

It is not possible to fully separate the effect of the initial ozone chemistry on the OH chemistry in the reactor. However, it is possible to measure the SOA formation caused by ozone only by mixing the ozone with the sample with UV lamps turned off.

4) Page 6, line 19-20: Although the authors have shown that the assumption that sulfuric acid losses to the reactor wall are negligible, it is not clear whether this is meant to suggest that such assumption would also hold for oxidation products of organic compounds, which have a wider range of volatility distribution. This should be discussed in the manuscript as it represents a limitation on the ability to quantify SOA yield. On a related note, the sulfuric acid yield section (3.4) is too short to stand alone as it is. The discussion need to be expanded to address this comment.

This is an important aspect that was pointed out by Referee #1, too. The losses of organic compounds are now discussed in Sect. 2.5 and Sect. 3.4. As a result, the sulfuric acid yield section is not standalone anymore. According to the model described in Sect. 2.5, the losses of low-volatility organics in toluene yield measurements (Sect. 3.5) in the TSAR are in similar level as in the PAM.

5) Page 7, line 9: It is not clear why the authors did not use the PAM reactor in the OFR254 mode given intended purpose of comparing the results with TSAR which uses a 254nm light source. This should be explained and justified. In addition, the manuscript provides no discussion of the potential effect of the different light sources on the non-OH chemistry in the reactors. The authors should include such discussion in the manuscript in the context of the work published by Peng et al., (2016) quantifying the extent of OH vs. non-OH chemistry according to the conditions applied in the reactor. It is important to understand the role of water mixing ratio, photon flux and external OH reactivity in the experiments on non-OH chemistry (photolysis) in order to establish the atmospheric relevance of the experiments.

See response to Referee #1 (comment 6) concerning the choice of OFR185 for the PAM.

The photolysis is now discussed in sections 2.5 and 3.4 by referring to results in Peng et al. (2016).

**6) Page 11, line 21: What is the background mass of the TSAR? How variable is it depending on OH and humidity conditions? Is this characterised and corrected for on a regular basis? More discussion of this should be included.**

The background mass is not a feature of the TSAR, but appears also in the PAM and results from the oxidation of the impurities in the dilution air. The dilution air should be cleaned as well as possible to avoid this artefact (this applies to smog chamber experiments as well). The purity of the air is always checked by oxidizing the pressurized air and measuring the mass formed. We expect the dependence on OH exposure and humidity is similar to that of secondary aerosol formation in general. This discussion and the values for the background mass are now added to Sect. 3.5 (p. 18, line 14-17 in the new manuscript).

7) Page 11, line 25-28: More discussion is needed for the apparent link between shorter residence time and the smaller size distribution produced in the TSAR experiment. The potential implication of such phenomena on the produced SOA particles and their properties needs to also be discussed.

In Sect. 3.5, the SOA mass is formed via nucleation and condensation. In TSAR, the oxidation is faster, which leads to higher concentrations of low-volatility compounds. Thus, the ratio of nucleation rate to condensation rate may be higher in the TSAR than in the PAM (Eq. (5) in the new manuscript), which results in smaller particles (but higher concentration) than in the PAM. The modeling of nucleation is complicated, so the validity of this speculation remains a future task. The properties of the SOA particles are expected to be similar based on the results in Sect. 3.5.

The effect of short residence time in general is now discussed in Sect. 2.5 and 3.4, and the discussion in Sect. 3.5 is also extended (p. 18 line 19-26 in the new manuscript).

8) Page 12, line 1-2: How representative is the off-line OH exposure calibration of the actual reported OH exposure in a more complex VOC mixture such that found in the Toluene SOA experiment or other VOC mixtures with different OH reactivates? This is likely to be a source of significant uncertainty in the determination of OH exposure and it is not discussed adequately in the manuscript.

This is a good point. The off-line OH exposure is only the upper limit for the OH exposure in the toluene experiment, since the OH reactivity in the toluene sample is higher than in the off-line calibrations (Li et al., 2015; Peng et al., 2015). We improved the estimation of OH exposure in the toluene experiment by using a photochemical model (see response to Referee #1, comment 10).

This OH exposure estimation improves Sect. 3.5 significantly, and as a result there is also better agreement between the TSAR and the PAM than in the first version of the manuscript. The figures 8, 9 and 10 were replaced with new ones because of the change in the OH exposure values (Figs. 10, 11 and 12 in the new manuscript).

9) Page 12, line 9-15: The discussion of the different toluene SOA yields among the Kang et al., Ng et al., and Hilderbrandt et al., is very brief and over-simplified. There are so many factors affecting the different studies that could potentially contribute to the reported SOA yields and differences cannot be explained only be the presence or absence of seed particles.

The aim of the toluene experiments were to compare the TSAR results to the PAM results at similar conditions. Thus, the comparison to other experiments (at different conditions) is now removed.

10) Page 12, line 18: The Aiken et al., (2008) analysis method for HR-AMS data has been updated by Canagaratna et al., (2015), with the new method having a direct effect on the reported O:C values. The authors should either justify the reason why the opted to use the Aiken calibration or update the results using the Canagaratna method.

At the time of the data treatment, the possibility to use Canagaratna method was not available in the software. The O:C values are now calculated using the Canagaratna method and the new values are applied to Figs. 11 and 12 (even though the difference is minimal). The text was changed as follows: "Elemental ratios are calculated using the method developed by Aiken et al. (2008) and improved by Canagaratna et al. (2015)."

**Editorial comments:**

Page 4, line 11: section should be 2.2 not 2.1 (same correction should be applied to all subsequent subsections in this part of the manuscript).

**The section numbering is now corrected.**

Page 10, line 14-16: the temperature and relative humidity should be reported as average with an associated standard deviation instead of the current mixing up of average with range.

2Finnish Meteorological Institute, Atmospheric Composition Research, P.O. Box 503, FI-00101 Helsinki, Finland 3VTT Technical Research Centre of Finland Ltd., P.O. Box 1000, FI-02044 Espoo, Finland

Correspondence to: pauli.simonen@tut.fi

**10 Abstract**

Oxidation flow reactors or environmental chambers can be used to estimate secondary aerosol formation potential of different emission sources. Emissions from anthropogenic sources, such as vehicles, often vary on short timescales. For example, to identify the vehicle driving conditions that lead to high potential secondary aerosol emissions, rapid oxidation of exhaust is needed. However, the residence times in environmental chambers and in most oxidation flow reactors are too long to study

- 15 these transient effects (~100 s in flow reactors and several hours in environmental chambers). Here, we present a new oxidation flow reactor, TSAR (TUT Secondary Aerosol Reactor), which has a short residence time (~40 s) and near-laminar flow conditions. These improvements are achieved by reducing the reactor radius and volume. This allows studying e.g. the effect of vehicle driving conditions on secondary aerosol formation potential of the exhaust. We show that the flow pattern in TSAR is nearly laminar and particle losses are negligible. The secondary organic aerosol (SOA) produced in TSAR has a similar
- 20 mass spectrum as the SOA produced in the state-of-the-art reactor, PAM (Potential Aerosol Mass). Both reactors produce the same amount of mass, but the TSAR has a higher time-resolution. We also show that the TSAR is capable of measuring secondary aerosol formation potential of a vehicle during a transient driving cycle, and that the fast response of the TSAR reveals how different driving conditions affect the amount of formed secondary aerosol. Thus, the TSAR can be used to study rapidly changing emission sources, especially the vehicular emissions during transient driving.

25

**1** Introduction**

[revised manuscript text omitted]

peak of SOA formed in PAM, whereas the concentration of total hydrocarbons measured from the exhaust shows much more transient behavior. Thus, the SOA formed in PAM cannot be linked directly to the emissions. To address this limitation, the residence time must be shortened. An ultimate example of a short-residence time flow reactor is the micro-smog chamber (MSC) with residence time < 10 s (Keller and Burtscher, 2012). However, (Bruns et al.,-(2015) show that the composition and amount of the SOA produced in MSC usually differs from those of the SOA produced in PAM or in an environmental chamber, possibly because of insufficient time for condensation of oxidant products. Thus, a compromise between PAM and MSC is needed for studying rapidly changing emissions: shorter residence time than in PAM but still long enough to allow the

5 condensation on aerosol phase.

In this work, we introduce and present a characterization of a new oxidation flow reactor, the *TUT Secondary Aerosol Reactor* (*TSAR*). The TSAR is better suited to measuring the real-time secondary aerosol formation potential of rapidly changing emission sources than the state-of-the-art oxidation flow reactors due to its improved flow conditions and shorter residence

- 10 time. In the following sections, we characterize the TSAR by describing its particle losses, oxidant exposure, residence time distribution, laboratory studies on sulfur acid yield as well as toluene SOA yield and properties, including a comparison between PAM and TSAR. In addition, we present measurements of the secondary aerosol formation of gasoline vehicle emissions during a transient driving cycle. We show that the fast response of TSAR gives valuable information on the effect of the driving condition on secondary aerosol formation potential.
- 15

30

Because of the high oxidant concentrations, high UV light intensity at non-tropospheric wavelengths and limited time for the condensation, atmospheric implications cannot be directly drawn from flow reactor measurements. However, there are no methods to measure the absolute secondary aerosol formation potential, because the environmental chambers also have their drawbacks (e.g. limited oxidant exposure and inability to measure time-resolved secondary aerosol potential) (Bruns et al.,

20 2015). Despite these artifacts, there is a need for estimation of secondary aerosol formation from different emission sources. Thus, also the flow reactor results provide useful information, on condition that a proper error analysis is made. In this work, we address the flow-reactor related artefacts of TSAR by modeling the vapor losses caused by photolysis and the short residence time.

**2** Experimental**

**25 2.1 Oxidation flow reactor**

TSAR is an OFR-254 type oxidation flow reactor, according to terminology proposed by Li et al. (2015), which means that OH radicals are produced from the photolysis of the ozone at 254 nm UV radiation. Its layout is presented in Fig. 1 (see Fig. S3 for a photograph). The TSAR consists of a residence time chamber (1 in Fig 1), an oxidation reactor (3), an ozone generator, three mass flow controllers and an expansion tube (2) that connects the residence time chamber and oxidation reactor. The residence time chamber is a 50 cm x 5 cm ID stainless steel cylinder that ensures the mixing of the sample and makes the sample flow laminar before entering the oxidation reactor. The half cone angle of the expansion tube is 6 degrees. Two of the

- mass flow controllers are connected to a vacuum line and are used to control the flow rates inside the residence time chamber and the oxidation reactor. The excess flow from the oxidation reactor is hereafter called "secondary excess flow". The third mass flow controller adjusts the air flow through the ozone generator. All the components except the residence time chamber
- 35 and the expansion tube are located inside a single housing, which makes TSAR easy to transfer to different measuring environments.

The TSAR oxidation reactor is a 3.3 l (52 cm x 9 cm inner diameter) quartz glass cylinder surrounded by two constant power ozone free low pressure mercury lamps which emit 254 nm UV light. The lamps are placed outside the reactor to ensure

40 laminar flow and to decrease the surface-to-volume ratio. The UV radiation generates excited oxygen atoms [O(1D)] from the photolysis of O3. These atoms react with water molecules, producing OH radicals. The O3 needed for this reaction chain is

mixed with the sample prior to the residence time chamber. In some cases, the humidity of the sample is too low for sufficient OH generation and additional humidification is required; in these cases, humidified air is also mixed into the sample at this point-. If the lamps emitted also 185 nm UV light, no external ozone generator would be needed and the TSAR would operate in OFR185 mode. However, we chose the OFR254 mode because of the poor transmission efficiency of the quartz glass for

5 185 nm light. In addition, the 185 nm light would generate ozone in the room air, which would require a special ventilation for the TSAR casing to avoid health issues.

The ozone is generated by an external ozone generator (either Model 600 or Model 1000, Jelight Company Inc.), which produces ozone from oxygen photolysis by 185 nm UV radiation. The ozone concentration can be adjusted by partially covering the UV lamp (Model 600) or by adjusting the flow rate through the generator.

The TSAR outlet is a 10 mm OD stainless steel probe, and its axial position can be adjusted, so that the oxidized sample can be measured from any distance from the inlet. From the probe, the sample is led to the measurement devices or to an ejector diluter, which allows the use of multiple instruments while maintaining a constant flow through the oxidation reactor.

**15 2.21 Residence time distribution experiments**

The flow conditions inside the TSAR oxidation reactor affect the dynamic transfer function E(t) of the reactor for non-reacting compounds. For this case, the measured temporal output concentration  $C_{out}(t)$  of the TSAR for a measured dynamic input concentration  $C_{in}(t)$  is the convolution of the measured input concentration and the transfer function (Fogler, 2006):

$$C_{out}(t) = E(t) * C_{in}(t)$$
(1)

20

10

The transfer function E(t) also is the unit impulse response of the reactor or the residence time distribution following an ideal Dirac delta input impulse. To test the response function, ten-second square pulses of CO2 were injected into the TSAR mixed with pressurized air. To keep the shape of the CO2 pulse as sharp as possible, the volumetric flow rate in the residence time chamber was kept at 50 slpm. In the oxidation reactor the flow rate was 5 slpm. CO2 concentration was measured with a CO2

analyzer (Sidor, Sick Maihak). As the same instrument is used for the measurement of both input and output concentrations, its response function is imbedded both in  $C_{in}(t)$  and  $C_{out}(t)$ .

First, three separate  $CO_2$  pulses were measured with sampling at the end of the residence time chamber. The outlet probe was then adjusted to sample at the end of the oxidation reactor and three separate pulses were again measured. The residence time distributions were determined for different situations: UV lamps and the secondary excess flow were either on or off.

**2.32 Particle loss quantification**

Particle losses in the oxidation reactor were measured using dioctyl sebacate (DOS) particles with mobility diameter from 20 to 100 nm and silver particles from 5 to 30 nm. The DOS particles were generated by atomizing DOS-isopropanol solution. The silver particles were generated with an evaporation-condensation technique (Harra et al., 2012). In these experiments, the volumetric flow in both the residence time chamber and oxidation reactor was 5 slpm.

35

30

A narrow monodisperse particle size distribution, size-selected using a differential mobility analyzer (nano-DMA, TSI Inc. Model 3085), was injected into the TSAR. The particle number concentration was measured with an ultrafine condensation particle counter (UCPC, TSI Inc. Model 3025) before and after the oxidation reactor using the adjustable outlet probe. This

40 procedure was repeated two or three times for each particle size.

**2.4 OH exposure experiments**

15

The length of the duration of atmospheric oxidation that the oxidation flow reactor simulates is determined by exposure of the sample to OH radicals. OH exposure (OHexp) is defined as  $[OH] \times t$ , where [OH] is the mean OH radical concentration in the oxidation reactor and t is the mean residence time of sample in the reactor. OHexp could be measured indirectly by monitoring

5 the loss of SO2 in the reactor.(Lambe et al., 2011) Since the only significant loss of SO2 in the oxidation reactor is due to the reaction with OH radicals (and possible wall loss), the change in SO2 concentration is defined by the following differential equation:

$$\frac{d[So_2]}{dt} = -k_{OH+SO_2} [OH] [SO_2] - k_{wall} [SO_2], \tag{2}$$

10 where  $[SO_2]$  is the SO2 concentration, and  $k_{OH+SO_2}$  is the reaction rate constant and  $k_{wall}$  is the first-order wall loss for SO2. From this, we get the OH exposure

$$OH_{exp} = \frac{1}{k_{OH+SO_2}} \ln \frac{[SO_2]_0}{[SO_2]_f},$$
(3)

where  $[SO_2]_0$  and  $[SO_2]_f$  are the SO2 concentrations of the sample before and after oxidation, respectively. Because both  $[SO_2]_0$  and  $[SO_2]_f$  are measured after the TSAR, the first without UV lights and the latter with UV lights, the wall loss term cancels out from the equation.

Because the OH radicals are produced in a reaction between water molecules and O(1D) atoms produced by ozone photolysis, both humidity and ozone concentration affect the amount of OH radicals (Seinfeld and Pandis, 1998). OHexp was measured using three different relative humidities (15 %, 30 % and 45 %) and several different ozone concentrations (0.6 ppm–49 ppm).
Pressurized air, SO2, Hhumidified air, and ozone and SO2 diluted with pressurized air were injected into the TSAR to determine the OH exposure. First, humidity, ozone concentration and [SO2]0 were measured after the TSAR. Then the UV lamps were turned on, and the concentration rapidly decreased and stabilized to the value of [SO2]f. SO2 concentration was measured with AF22M analyzer (Environnement S.A) and ozone with model 205 analyzer (2B Technologies).

25 Based on the OHexp measurements, it is possible to deduce the UV actinic flux in the TSAR by reproducing the results in a photochemical model and using the photon flux as a fitting parameter. We used the model available in PAM users manual (PAM\_chem\_v8 by William Brune, https://sites.google.com/site/pamusersmanual/7-pam-photochemistry-model/a-introduction), which is similar to the model described by (Li et al.,-(2015). In this model, the differential equations describing the chemical reactions are solved using Euler's method (instead of Runge-Kutta method used in the model by (Li et al.,-(2015)).

**30 2.5 Estimating vapor losses and photolysisSulfuric acid yield experiments**

In an ideal oxidation flow reactor, all the condensable vapors condense onto particle phase and will be measured as potential secondary aerosol mass. However, there are also other pathways than condensation for the vapors in the flow reactor, and some of them are non-tropospheric. First, the intensity of the UV radiation is higher and the wavelength is smaller than those of the UV radiation in the troposphere. This can cause unrealistic photolysis of the precursor vapors and the secondary aerosol formed

- 35 (Peng et al., 2016). Second, the residence time in the flow reactor is small, and thus the condensable vapors may exit the reactor before condensing onto particle phase. Third, because of high oxidant concentrations, the timescale of condensation can be much higher than the timescale of oxidation, leading to fragmentation of oxidized vapor molecules before they have condensed. This is of concern especially in the TSAR, where the short residence time requires higher oxidant concentrations than e.g. PAM chamber. Fourth, the surface-area-to-volume ratio is high in the flow reactor, and thus the vapor wall losses may be
- 40 significant.(Palm et al., 2016)

**2.5.1 Photolysis**

(Peng et al., (2016) have studied the losses of precursor gases and SOA due to photolysis in flow reactors. In their study, they show that the photolysis rate of SOA in oxidation flow reactors is uncertain because of the lack of knowledge on quantum yields. In any case, the loss of SOA due to photolysis is much smaller in oxidation flow reactors than in the troposphere at

5 equivalent OH exposure. However, the photolytic losses of precursor gases in oxidation flow reactors can be higher than in the troposphere.

The photolytic loss is significant if the photolysis rate is high relative to reaction rate with OH radical. We define relative photolytic loss as

10 relative photolytic loss =  $\frac{photolysis rate at 254 nm}{photolysis rate at 254 nm+reaction rate with OH}$  (4)

A relative photolytic loss of zero means that all the loss of the precursor gas is due to reaction with OH, and relative photolytic loss of unity means that the photolysis is the only pathway of loss for the precursor gas. As Peng et al. (2016) show, the relative photolytic loss depends on the ratio of photon exposure to OH exposure ( $F254_{exp}/OH_{exp}$ ), the reaction rate constant between

- 15 the precursor molecule and OH radical, the absorption cross section of the molecule and the quantum yield of the photolysis reaction. The OH exposure in the TSAR depends on water vapor concentration ([H2O]), ozone concentration and external OH reactivity of the sample (OHRext =  $[X] \cdot k_{OH+X}$ , where [X] is the precursor gas concentration and  $k_{OH+X}$  is the reaction rate constant between this gas molecule and OH radical). The photon flux in TSAR is constant.
- According to the modeling results by Peng et al. (2016), the relative photolytic loss of studied precursor gases is less than 60 % in most cases in OFR254, even at "riskier" conditions ([H2O] < 0.1 % or OHRext > 200 s-1). For most of the studied precursor gases, the relative photolytic loss is less than 20 % in most cases. In all the studied "safer" conditions ([H2O] > 0.5 % and OHRext < 50 s-1), the relative photolytic loss is less than 30 % for all the precursor gases. However, these are only the upper limits for the relative photolytic losses because of the assumption of a unity quantum yield. The relative photolytic losses in
- 25 TSAR are discussed in Sect. 3.4.1.

**2.5.2 Vapor losses**

We study the fate of condensable vapors (other than photolysis) in TSAR using a similar approach as (Palm et al., (2016).We start with a low-volatile organic compound (LVOC, saturation vapor concentration ~0) which can condense on particle phase, condense on the reactor walls, form new particles via nucleation, react with OH radical or exit the reactor before condensing.

| 30 | Thus, the concentration of the LVOC is described with the following differential equation                                                                                   |
|----|-----------------------------------------------------------------------------------------------------------------------------------------------------------------------------|
|    | $\frac{dC_0}{dt} = -4\pi \cdot D \cdot CS \cdot C_0 - k_w \cdot C_0 - k_{OH} \cdot C_0 \cdot [OH] - n \cdot J(C_0), $ (5)                                                   |
|    | where $C_0$ is the concentration of the initial LVOC, $D$ is the diffusion coefficient of the LVOC, $CS$ is the condensational sink,                                        |
|    | $k_{w}$ is first-order rate coefficient for wall-loss, $k_{OH}$ is the reaction rate constant between OH radicals and the LVOC, [OH] is                                     |
|    | the mean concentration of OH radicals in the reactor, n is the number of molecules in a nucleated particle and J is the                                                     |
| 35 | nucleation rate, which depends on the vapor concentration. We assume that the reaction with OH radical produces another                                                     |
|    | LVOC (<math>C_1</math>) which has the same loss terms as <math>C_0</math>. Thus,                                                                                     |
|    | $\frac{dC_1}{dt} = k_{OH} \cdot C_0 \cdot \left[OH\right] - 4\pi \cdot D \cdot CS \cdot C_1 - k_w \cdot C_1 - k_{OH} \cdot C_1 \cdot \left[OH\right] - n \cdot J(C_1) $ (6) |
|    | and more generally,                                                                                                                                                         |
|    | $\frac{dC_n}{dt} = k_{OH} \cdot C_{n-1} \cdot [OH] - 4\pi \cdot D \cdot CS \cdot C_n - k_w \cdot C_n - k_{OH} \cdot C_n \cdot [OH] - n \cdot J(C_n), $ (7)                  |

40 assuming that  $k_{OH}$ , D, CS,  $k_w$ , n and J are equal for all oxidation products.

At some point, the reaction between LVOC and OH radical leads to fragmentation and produces high-volatility compounds which cannot condense onto particle phase. Palm et al. (2016) assumed that the fifth oxidation reaction produces fragmented compounds. In addition, the heterogeneous OH reaction on the particle surface may result in fragmentation (Kroll et al., 2009).

| 5  | Thus, assuming that the molecule fragments to two parts, we get                                                                     |                    |  |  |  |  |  |
|----|-------------------------------------------------------------------------------------------------------------------------------------|--------------------|--|--|--|--|--|
|    | $\frac{dC_5}{dt} = 2 \cdot k_{OH} \cdot C_4 \cdot [OH] + 2 \cdot R_{heterogeneous}$                                                 | (8)                |  |  |  |  |  |
|    | where $C_5$ is the mass concentration of fragmented, high-volatility compounds and $R_{heterogeneous}$ is the rate of heterogeneous |                    |  |  |  |  |  |
|    | fragmentation. Based on these equations, the molecule flux to the aerosol phase is                                                  |                    |  |  |  |  |  |
|    | $\frac{dC_{aer}}{dt} = 4\pi \cdot D \cdot CS \cdot (C_0 + C_1 + C_2 + C_3 + C_4) - R_{heterogeneous}$                               | (9)                |  |  |  |  |  |
| 10 | and the mass flux to the reactor walls is                                                                                           |                    |  |  |  |  |  |
|    | $\frac{dc_w}{dt} = k_w \cdot (C_0 + C_1 + C_2 + C_3 + C_4).$                                                                        | (10)               |  |  |  |  |  |
|    |                                                                                                                                     |                    |  |  |  |  |  |
|    | The fate of LVOC is obtained by solving the differential equations using MATLAB (Release 2016a, The MathWo                          | orks, Inc., |  |  |  |  |  |
|    | United States) ode45 numerical solver. The fraction of LVOC lost to walls is then                                                   |                    |  |  |  |  |  |
| 15 | $F_{wall} = \frac{C_w(\tau_{res})}{C_0(0)},$                                                                                        | (11)               |  |  |  |  |  |
|    | where $\tau_{res}$ is the residence time of the reactor and $C_0(0)$ is the initial LVOC concentration. Similarly, the fraction     | of LVOC            |  |  |  |  |  |
|    | condensed onto aerosol is                                                                                                           |                    |  |  |  |  |  |
|    |                                                                                                                                     |                    |  |  |  |  |  |

$$F_{aer} = \frac{c_{aer}(c_{res})}{c_0(0)},\tag{12}$$

(13)

(14)

the fragmented fraction is

20
$$F_{frag} = \frac{1}{2} \frac{c_5(\tau_{res})}{c_0(0)}$$
,
and finally, the fraction of LVOCs that exit the reactor before condensing is

$$F_{exit} = \frac{C_0(\tau_{res}) + C_1(\tau_{res}) + C_2(\tau_{res}) + C_3(\tau_{res}) + C_4(\tau_{res})}{C_0(0)}$$

- Looking at Eqs. (5-8), the OH radical concentration affects the relative amount of LVOC that is fragmented. The shorter the residence time, the higher the [OH] must be to attain a certain OH exposure. Thus, shortening the residence time results in increase in fragmented LVOC. However, the fragmented fraction depends on the timescales of the other loss terms, namely condensation on reactor walls and on aerosol that in turn depends on the condensational sink. Using this approach, the dependence of LVOC fate on residence time and condensational sink are studied in Sect 3.4.2.
- 30 We tested the model validity by oxidizing  $SO_2$  in the TSAR.  $SO_2$  oxidation is a simple example of secondary aerosol formation. SO2 reacts with OH radicals to produce sulfuric acid (H2SO4) vapor which rapidly enters the particle phase by nucleation and condensation (Sihto et al., 2006). The mass formed by oxidation of SO2 can be theoretically calculated from the SO2 loss, and thus comparing the measured mass formation to the theoretical prediction can be used to estimate the capability of TSAR to simulate full atmospheric oxidation. Should the measured mass be substantially smaller than the theoretical, we would assume
- 35 that there are significant losses of sulfuric acid vapor inside TSAR. This would then also mean that the losses for organic lowvolatile and semi-volatile vapors were also high. The observed losses can then be compared to the losses predicted by Eqs. (11-14).

The sulfuric acid yield was measured by injecting-pressurized air,  $SO_{27}$ , humidified air-and, ozone and  $SO_2$  diluted with pressurized air into the TSAR. The relative humidity and  $SO_2$  was measured straight after TSAR, whereas ozone concentration and the particle size distribution were measured after an ejector dilutor (Dekati Ltd.). The dilution ratio was determined by measuring the sample flow rate and the dilution air flow rate. The particle size distribution was measured with a nano-SMPS (scanning mobility particle sizer: a nano-DMA (TSI Inc. model 3085) combined with a UCPC (TSI Inc. model 3025)).

5 In addition to sulfuric acid, the measured particles also contain water. The sulfuric acid mass was calculated from Eq. (415) (Lambe et al., 2011).

 $m_{H_2SO_4} = \chi_{H_2SO_4} \times V \times \rho, \tag{415}$

where  $\chi_{H_2SO_4}$  is the mass fraction of sulfuric acid in the particle phase, *V* is the volume calculated from nano-SMPS particle size distribution and  $\rho$  is the density of the particle phase. Both the mass fraction and the density were calculated as a function

10 of relative humidity based on Seinfeld and Pandis (1998). In the calculations, relative humidity after the dilution is used, assuming fast equilibration of the sulfuric acid particles.

The theoretical (maximum) sulfuric acid mass was calculated by multiplying the loss of  $SO_2$  by the molar mass of a sulfuric acid molecule. Thus, the loss of 1 ppb of  $SO_2$  produces 4.03 µg m-3 of sulfuric acid aerosol, assuming also that all the sulfuric acid condenses into the particle phase.

**2.6 Organic precursor experiments**

A key application of TSAR is to estimate the amount of secondary aerosol mass formed from engine exhaust emissions, which in turn contains a complex mixture of organic and inorganic gases. Therefore, the  $SO_2$  oxidation experiment alone is not a representative example of engine exhaust oxidation, because the oxidation pathways of organic compounds are far more

20 complex. The ability of TSAR to form SOA was verified by measuring the toluene SOA obtained by TSAR and the PAM simultaneously. Previous studies have shown that the amount and properties of the SOA produced in the PAM are similar to those of the SOA formed in smog chambers, which represent atmospheric oxidation (Bruns et al., 2015; Lambe et al., 2015).

The organic precursor gas in this experiment was toluene, because it is present in engine exhaust gas (Peng et al., 2012; Wang et al., 2013). In addition, toluene is globally one of the most emitted anthropogenic SOA precursors (Kanakidou et al., 2005). Gas-phase toluene was produced using a permeation oven with a toluene permeation tube (KIN-TEK Laboratories Inc.), and its output rate ( $\dot{M}_{toluene}$ ) was measured by weighing the change in its mass. The concentration of toluene in the reactors is

$$C_{toluene} = \frac{\dot{M}_{toluene}}{Q_{tot}},\tag{516}$$

where  $Q_{tot}$  is the total sample flow through the reactors (10 slpm).

30

15

The gas-phase toluene was mixed with ozone and humidified air before it was fed to the TSAR residence time chamber. After the residence time chamber, 5 slpm of the sample was introduced into the TSAR oxidation reactor and 5 slpm to the PAM. A 4-way valve was installed after the reactors, so that the instruments were sampling from one reactor while the sample from the other reactor was drawn to the vacuum line through a mass flow controller.

The PAM was used in OFR185 mode (Li et al., 2015) and thus the external ozone generator was switched off when the instruments were sampling from the PAM. The PAM was operated in OFR185 mode instead of OFR254 mode because the OFR185 mode is used in previous engine exhaust studies (Karjalainen et al., 2016; Timonen et al., 2016; Tkacik et al., 2014). Similar results from the two reactors would then indicate that the TSAR operating in OFR254 mode can be used in similar

[revised manuscript text omitted]

The measurement results could be reproduced in the photochemical model using the photon flux, first-order OH radical wall

5 loss and first-order ozone wall loss as free parameters. The best fit values are  $1.92 \times 10^{15}$  photons cm-2 s-1 (254 nm photon flux), 8.3 s-1 (OH wall loss) and 7.5 ×  $10^{-4}$  s-1 (ozone wall loss). Using these parameters, the model predicts the measurement results within ±20 % uncertainty when the relative humidity, temperature, initial ozone concentration and initial SO2 concentration are used as the input parameters (Fig. S4).

**3.4 Sulfurie acid yield Vapor losses and photolysis in TSAR**

**10 3.4.1 Photolysis**

Based on the modeling results in Sect. 3.3, the flux of 254 nm photons in TSAR is **1.92** × **10**15 photons cm-2 s-1 and does not depend on OH exposure, since OH exposure is adjusted by O3 and H2O concentration. Assuming residence time of 37 seconds (Sect. 3.1), the ratio  $F254_{exp}/OH_{exp}$  is shown in Fig. 6a. When the  $OH_{exp} > 10^{11}$  molec. cm-3 s (~0.8 days equivalent atmospheric exposure), the ratio is less than 106 cm s-1. According to Peng et al. (2016), the relative photolytic loss for most VOCs is below

- 15 20 % at this ratio. The only exceptions are acetylacetone, E,E,2,4-hexadienedial, peroxacetyl nitrate and species with multiple hydroxyls and carbonyls, whose relative photolytic loss is 30-60 % when F254exp/OHexp is 106. At higher OHexp, the relative photolytic losses decrease. Thus, to avoid non-tropospheric photolysis of precursor gases, the OH exposure must be maintained high enough. However, we note that these relative photolytic losses are upper limits, since a unit quantum yield is assumed in the calculations.
- 20

As shown by (Peng et al., (2015), the OHexp in OFR254 depends on water vapor concentration, OHRext, photon flux and ozone concentration. Using the photochemical model described in Sect. 2.4, we evaluate the effect of OHRext on F254exp/OHexp while keeping the temperature, relative humidity and initial ozone concentration constants (20 °C, 30 % and 45 ppm, respectively). According to the model results (Fig. 6b), F254exp/OHexp < 106 cm s-1 as long as OHRext < 2500 s-1. On the other hand, the OHexp

25 decreases as a function of  $OHR_{ext}$ . A worst-case scenario regarding the  $OHR_{ext}$  in exhaust measurements is a cold engine start, where the  $OHR_{ext}$  can be as high as  $1000 - 3400 \text{ s}^{-1}$  (excluding the effect of NOX) when the dilution ratio is ~12 (Karjalainen et al., 2016; Timonen et al., 2016). The reactions between NOX, O3 and OH also decrease the  $OH_{exp}$  and therefore increase the photolysis rate of VOCs. Thus, a higher dilution ratio than 12 should be used when sampling cold-start engine exhaust into the TSAR.

**30 3.4.2 Vapor losses**

To study the dependence of LVOC fate on residence time and condensational sink, we define two cases: the oxidation of ambient air (low condensational sink) and diluted vehicle exhaust (high condensational sink). The condensational sink depends on particle number concentration and size, and also on the accommodation coefficient ( $\alpha$ ), diffusion coefficient and molecular mass of the condensing vapor. The first-order rate coefficient for wall-loss ( $k_w$ ) is calculated as in Palm et al. (2016) (see Supplementary material for details on calculation of CS and  $k_w$ ). Following the example in Palm et al. (2016), we assume the following properties for the LVOC: molar mass of 200 g mol-1, diffusion coefficient (D) of  $\mathbf{7} \times \mathbf{10}^{-6}$  m2s-1, and  $k_{OH} = \mathbf{1} \times \mathbf{10}^{-11}$  cm3 molec.-1 s-1. For simplicity, we assume that the nucleation rate in Eqs. (5-7) is zero. Nucleation is still implicitly taken into account because the condensational sink is calculated from average size distribution before and after TSAR. In addition, we do not consider the heterogeneous fragmentation.

In the case of oxidation of ambient air, the timescale for condensation on aerosol  $\tau_{aer} = (4\pi \cdot D \cdot CS)^{-1} \approx 65 \text{ s}$  (when  $\alpha = 1$ ). According to Palm et al. (2016), this is a typical value for ambient pine forest air oxidized in PAM chamber, when sufficient amount of precursors are available for SOA formation. This CS is equivalent to that of a log-normal particle size distribution with total number concentration of  $1.5 \times 10^5$  cm-3, median diameter ( $\mu$ ) of 25 nm and geometric standard deviation ( $\sigma$ ) of 1.4.

5 The use of particle size distribution instead of a constant CS allows us to vary  $\alpha$  (since CS depends on  $\alpha$ ). Different values for LVOC mass accommodation coefficients have been proposed. For example, (Saleh et al., (2013) measured a value of  $\alpha \approx 0.1$  for alpha-pinene SOA, whereas Palm et al. (2016) argue that  $\alpha \approx 1.0$  for ambient pine forest SOA.

The vapor losses in TSAR for the ambient case as a function of residence time were modeled using the method described in Sect. 2.5.2. Two values of the mass accommodation coefficients were used ( $\alpha = 0.1$  and  $\alpha = 1.0$ ). The equivalent OHexp is 5 days regardless of the residence time. The results are presented in Fig. 7a. At typical TSAR residence time (37 s), the LVOC losses in this case are 69-96 %, depending on the value of  $\alpha$ . Most losses are caused by fragmentation, and longer residence time results in less losses (Fig. S5). This is because the shorter the residence time, the higher the OH concentration must be to reach the same equivalent OH exposure. When the OH concentration is high enough, the timescale of fragmentation is lower than that of condensation

15 than that of condensation.

The vehicle exhaust case is based on the measurements in Sect. 3.6. The mass concentration and  $\tau_{aer}$  of diluted primary aerosol are approximately 4.8 µg m-3 and 81 s-1, respectively (when  $\alpha = 1$ ). This is approximated as a log-normal size distribution with  $\mu = 31$  nm,  $\sigma = 1.9$  and number concentration of **5.4** × 104 cm-3. According to the measurements in Sect. 3.6, the mass concentration after TSAR is approximately 156 µg m-3 (when background is subtracted). For simplicity, we assume that the

- 20 concentration after TSAR is approximately 156  $\mu$ g m-3 (when background is subtracted). For simplicity, we assume that the increase in mass is caused only by condensation so that number concentration and  $\sigma$  are constant. Thus, the size distribution after TSAR is otherwise similar to the primary size distribution, but  $\mu = 103$  nm. The average CS in TSAR is calculated from the average of these two size distributions.
- 25 The results for the car exhaust case are presented in Fig. 7b. Now, the LVOC losses in TSAR at typical residence time are 25-80 % depending on the value of  $\alpha$ . The losses are lower than in the ambient case because of shorter timescale of condensation caused by the higher CS. Again, the highest loss is caused by fragmentation (Fig. S5).

In addition to TSAR, we present the estimates for LVOC losses in several other flow reactors, namely MSC, PAM reactor, and Caltech Photooxidation Flow Tube Reactor (CPOT) (Huang et al., 2016) at their typical residence times in Fig. 7. The results differ a little from the TSAR curve because of the different surface-area-to-volume ratios. One must note that the applications of the flow reactors are different; e.g. the MSC is usually used with much higher CS than what is modeled here (e.g. Corbin et al., 2015) and consequently the losses are smaller than in Fig. 7. Similarly, the main application of the TSAR is engine exhaust measurement, where the CS is usually higher than in ambient air.

35

40

The model is tested by comparing the measured and modeled sulfuric acid losses, and the results are shown in Fig. 8. In the model, the following values are assumed for the sulfuric acid molecules: molar mass of 98 g mol-1,  $\alpha = 0.65$  (Pöschl et al., 1998) and D = 1 × 10-5 m2 s-1(Hanson and Eisele, 2000; Palm et al., 2016). We assume there is no fragmentation for sulfuric acid molecules. The CS is again calculated from the average of size distributions after and before TSAR (in this case the size distribution after TSAR divided by two). For the three measurements with the smallest error bars, the measured sulfuric acid

loss is on average 4 %. The modeled loss, in contrast, is 18 % on average. The reason for this discrepancy may be the underestimated CS, since dividing the measured size distribution by two does not necessarily represent the average size distribution in TSAR. If instead the measured size distribution is used for CS calculation (the upper limit for average CS), the

model results in average loss of 6 %, which is much closer to the measured one and indicates that the nucleated particles generate a high CS already during the first steps of oxidation. Thus, the modeled losses for the ambient and vehicle exhaust case are probably slightly overestimated.

- 5 The sulfuric acid experiment shows that the model predicts the losses of a non-fragmenting low-volatile compound reasonably well. However, in Fig. S5 we see that it is the fragmentation that causes the highest losses for LVOCs when the residence time is short (< 50 s). The assumption that 5 oxidation steps result in fragmentation is artificial, but if we as a sensitivity test assume that the fragmentation does not occur at all, the change in overall loss is small because a higher proportion of the LVOCs will exit the reactor before condensing (Fig. S6). Still, the losses are little lower in case of no fragmentation, and thus more studies
- 10 on fragmentation are needed to verify the assumptions in the model.

The modeled cases inarguably show that there is a tradeoff between residence time and LVOC losses: the smaller the residence time, the more losses. Thus, the residence time must be chosen according to the application. If a short residence time is used and the CS is low, the injection of seed particles in the sample will reduce the LVOC losses. In the car exhaust case, the CS is

- 15 high enough for TSAR if the mass accommodation coefficient of the condensing vapor is close to unity. For steady-state experiments, we recommend using a long residence time when there is no need for fast response. However, even though the LVOC losses are smallest for long-residence time reactors according to the model results, the particle losses are higher (e.g. ~20 % for PAM and CPOT for 100 nm particles, in TSAR ~0%) (Huang et al., 2016; Lambe et al., 2011).
- 20 The measured sulfuric acid mass as a function of expected mass is shown in Fig. 6. For the three measurements with the smallest error bars, the measured mass is on average 4 % lower than the expected mass, indicating that there are no significant losses of sulfuric acid vapor or particles.

**3.5 Toluene SOA yield and properties**

The SOA formation studies were conducted as described in Sect. 2.6. Toluene concentration in the sample entering the reactors was 320 ppb ( $\pm$ 34 ppb). During the experiments, Thethe average temperature of sample was  $23 - 2423.6\pm0.2$  °C and the average relative humidity  $26 - 34.31.3\pm2.9$  %, where the uncertainty is the standard deviation of the values. with the exception of the PAM measurement at highest OH exposure, where the average relative humidity was 37 %.

**3.5.1 Steady state experiments**

The SOA mass formed in the reactors is calculated from the number size distribution measured by the SMPS, assuming
spherical particles with a density of 1.45 g cm-3 (Ng et al. 2007). The SMPS was used for PM concentration measurements instead of the AMS because especially for the TSAR, all particles do not fall in the AMS detection range (40 nm–800 nm). The background mass, i.e. the mass formed in the reactors in the absence of toluene, was subtracted from the toluene SOA mass. The background mass consists of oxidation products of the dilution air, and depends on the purity of the pressurized air. The purity of the air must be always checked by measuring the mass formed in the flow reactor in absence of any exhaust or precursors. In this experiment, the background mass concentration was on average 1.2 µg m-3 for the TSAR and 1.1 µg m-3 for

the PAM. For comparison, the average mass concentration in toluene measurements was  $137 \,\mu g \,m^{-3}$ .

Figure 7-9 shows SMPS mass distributions of toluene SOA for PAM and TSAR at OHexp of 1.33.6× 10112 molec. s cm-3 and 6.01.1 × 10121 molec. s cm-3, respectively. The PAM produces a wide mass distribution where particles above 100 nm
contribute to approximately half of the total mass. The TSAR produces a narrower mass distribution where approximately half of total mass is located in particles smaller than 40 nm. This phenomenon was also reported by Bruns et al. (2015): the microsmog chamber, which is smaller, has a shorter residence time, and generates smaller particles than PAM. As discussed in sections 2.5.2 and 3.4.2, the shorter residence time limits the condensational growth of particles and may favor nucleation instead of condensation. Implementing nucleation in the LVOC fate model remains a future task, but an estimation of the losses of toluene oxidation products was conducted in a similar way as in Sect. 3.4.2. If the accommodation coefficient of the

- 5 oxidation products is one, the LVOC losses for the toluene SOA cases are less than 2 % except for TSAR measurement at  $OH_{exp}$  of  $3 \times 10^{10}$  molec. s cm-3, where the losses are approximately 8 %. If the accommodation coefficient is 0.1, the losses are less than 35 % except for the TSAR low  $OH_{exp}$  measurement, where the losses are approximately 73 %. Since the exact value of the accommodation coefficient is unknown, the results in this section are not corrected with the estimated losses.
- Figure 8-10 shows toluene SOA yield obtained in steady state experiments as a function of  $OH_{exp}$  for both reactors. The  $OH_{exp}$  was not measured simultaneously but is obtained as a function of  $O_3$  measured after the reactors from the model described in Sect. 2.4, assuming an  $OHR_{ext}$  equivalent to 320 ppb of toluene, and  $\pm 20$  % uncertainty in TSAR  $OH_{exp}$  and  $\pm 30$  % in PAM  $OH_{exp}$  (PAM model fitting results are presented in Fig. S7). Following the reasoning in (Peng et al., (2015), we use SO2 as a proxy for the toluene in the model, the off line calibrations. The maximum yield is approximately 0.2 for both reactors
- 15 (neglecting the TSAR outlier at  $OH_{exp}$  of  $1 \times 10^{12}$ -6.3 ×  $10^{11}$  molec. s cm-3 and uncertain PAM values at low  $OH_{exp}$ ), and both reactors reach the maximum yield at  $OH_{exp}$  between 0.35 ×  $10^{12}$  molec. s cm-3 and  $1.50 \times 10^{12}$  molec. s cm-3. However, the TSAR yield is more sensitive to  $OH_{exp}$  than the PAM yield. One possible reason is that because of the broad residence time distribution, the PAM always outputs SOA with both high and low  $OH_{exp}$  regardless of the mean  $OH_{exp}$ . This mixing of different aged SOA may stabilize the output and reduce the sensitivity to  $OH_{exp}$ . Even higher yields are observed in PAM at
- 20 low  $OH_{exp}$  (< 3 × 1011 molec. s cm-3), but the calculation of yield at this low  $OH_{exp}$  is very sensitive, because of the small value of the denominator in Eq. (17). This is seen as high uncertainty in PAM yields at low  $OH_{exp}$ . The uncertainty highlights the importance of simultaneous, accurate measurement of  $OH_{exp}$  especially when the PAM light intensity is low. At higher  $OH_{exp}$  the yields of the two reactors agree very well.
- 25 When the  $OH_{exp}$  is higher than  $1 \times 10^{12}$  molec. s cm-3, the yield starts to decrease in both reactors. This indicates that the assumption on fragmentation in the LVOC loss model (Sect. 2.5.2) is right: the higher the OH concentration, the more fragmentation occurs. The fragmented molecules will not condense on aerosol phase and thus the yield is lower. Still, these results cannot be used to validate the simplified model assumption that 5 oxidation steps of an LVOC leads to fragmentation.
- 30 The PAM yield in Fig. 10 is not corrected for particle losses, because the losses are characterized only for particles that enter the PAM at certain size and do not grow inside the PAM by condensation. This is not the case here, because the particles are formed via nucleation inside the PAM, and thus it is unknown how long they have spent in the reactor and what is the particle size as a function of residence time. As an estimate, the particle size-distribution measured after PAM was corrected with the losses measured for this particular chamber (Karjalainen et al., 2016). With this correction, the PAM yield would increase by
  35 10 % on average
- 35 19 % on average.

Kang et al. (2007) reported a toluene SOA yield of 0.09 using an early version of the PAM at similar relative humidity, temperature and toluene concentration as in this paper. Ng et al. (2007) and Hildebrandt et al. (2009) used smog chambers with seed particles and low relative humidity, and reported toluene SOA yields of 0.30 and 0.26 0.41, respectively. The SOA

40 yield in this study is therefore slightly lower than in smog chamber studies, which may be caused by the lack of seed particles: Lambe et al. (2015) showed that the addition of seed particles in the PAM resulted in a higher SOA yield, at least in the case of isoprene. In addition to yield, the chemical composition of produced SOA was studied. In Fig. 911, a van Krevelen diagram shows the oxidation state of SOA for both reactors. In this diagram, H/C ratio is shown as a function of O/C ratio. Elemental ratios are calculated using the method developed by Aiken et al. (2008) and improved by Canagaratna et al. (2015). Oxidation of aerosol usually increases the O/C ratio and decreases the H/C ratio (Heald et al., 2010). This phenomenon is observed in both reactors,

5 and based on these ratios, the oxidation state of SOA is similar in PAM and TSAR at comparable OH exposures. when the exposure is less than 7 equivalent days. At higher OHexp, similar ratios are still observed in both reactors, but at different OH exposures.

To further compare the SOA oxidation state in the reactors, the average carbon oxidation state ( $\overline{OS_c}$ ) of SOA is shown in Fig. 10 1012. The average carbon oxidation state is a metric which is invariant to hydration or dehydration, and is defined as  $\overline{OS_c} \approx$  $2 \times O/C - H/C$  (Canagaratna et al., 2015; Kroll et al., 2011). As well as the O/C ratios and H/C ratios, the  $\overline{OS_C}$  of the SOA in the reactors also agree at comparable OHexp. differ at high OHexp. The TSAR SOA has higher oxidation state than the PAM SOA even though the OHexp is similar. The trend of TSAR and PAM  $\overline{OS_c}$  as a function of OHexp seems to differ at higher

 $OH_{exp}$ , but this can be caused by the uncertainty in the  $OH_{exp}$  estimation, which is visualized with error bars in Fig. 12. 15

These discrepancies in the oxidation state suggest that the OHexp alone does not affect the composition of SOA. For example, in Fig. 10 the  $\overline{OS_c}$  of TSAR SOA increases from 0.32 to 0.63 even though the increase in OHexp is small (**2.0 × 10**11 molec. s cm-3). PAM SOA needs an increase of 4.0 × 1011 molec. s cm-3 in OHexp to achieve the same increase in  $\overline{OS_{cr}}$  (from 0.34 to 0.63). The difference between TSAR and PAM in the high  $\overline{OS_C}$  region is that the outlet ozone concentration in TSAR increases 20 by 42 ppm when OHexp increases by 5.1 × 1011 molec. s cm-2, whereas in PAM the outlet ozone concentration increases only by 7.2 ppm at comparable change in OHexp (6.4 × 1011 molec. s cm-3). Thus, the increasing ozone exposure in TSAR may lead to the increase in  $\overline{OS_{L}}$ , which suggests that the toluene oxidation products react with ozone. However, also the OHexp estimation may be uncertain, since it is based on off line calibration. In this experiment, the high concentration of toluene likely decreases the OHexp, and this effect may have a different magnitude between the reactors. This could also explain the discrepancies in Fig. 10.

We also compare T the chemical composition of SOA is further compared by studying the organic mass spectra. According to Marcolli et al. (2006) and Lambe et al. (2015), a dot product between two normalized mass spectra can be used to determine whether the spectra are similar. The spectra are normalized by dividing each signal by the square root of the sum of the squares 30 of all signals. A dot product of one implicates that the spectra are identical, and zero that they are orthogonal.

Toluene SOA here is divided in three categories: low oxidation (-0.18 <  $\overline{OS_C}$  < -0.16), medium oxidation (0.50 <  $\overline{OS_C}$  < 0.69) and high oxidation ( $\overline{OS_c} > 1.10$ ). The dot products between the organic spectra of different reactors are shown in Table 2. The dot products of normalized mass spectra of SOA produced in reactors at comparable  $\overline{OS_c}$  are above 0.99, indicating that the reactors produce similar SOA matter in regard to chemical composition.

The TSAR and PAM reactors differ in volume, geometry, flow conditions and residence time. The most significant difference is in the oxidation process: TSAR operates in OFR254 mode and PAM in OFR185 mode. However, the agreement between yields and organic mass spectra of SOA produced in both the TSAR and PAM reactors show that the oxidation products are similar in both reactors, at least in the case of toluene. In OFR254, the sample is first exposed to ozone (before the oxidation

reactor) and then to both ozone and OH radicals. If the VOCs in the sample react fast with ozone, resulting SOA mass might differ between OFR254 and OFR185. This was not the case for toluene, as dark experiments (only ozone, no UV light) did

25

35

not produce any secondary mass. In other applications, for example when oxidizing biogenic precursors which are highly reactive towards ozone, the results between OFR254 and OFR185 presumably differ, OFR185 being more realistic as the sample is exposed to ozone and OH simultaneously. However, the main application of TSAR is to measure vehicle emissions, which are more reactive towards OH than ozone (Gentner et al., 2012; Tkacik et al., 2014). The potential of ozone to produce SOA from the emission can be measured by injecting ozone into the TSAR with UV lights turned off.

**3.5.2** Pulse experiments**

5

20

25

The SOA mass concentrations as a function of time are shown in Fig. 11-13 for all pulse experiments. The 10 s pulse of toluene results in a sharp peak in mass in the TSAR, whereas the PAM reactor produces significantly broader peaks at both used flow rates. Interestingly, the TSAR mass peak is divided into two distinct peaks. We do not know the reason for this phenomenon
since the residence time distributions in Sect 3.1 do not support this kind of behavior. However, the flow conditions in this experiment are not exactly the same as in Sect 3.1: here, the flow rate in residence time chamber is only 10 slpm, whereas in Sect. 3.1 it was 50 slpm.

[revised manuscript text omitted]

- 35 factor over the cycle is the integral of the time-resolved emission factor over the cycle length divided by the total distance. For the primary emissions, the emission factor is 0.1 mg km-1 and for the secondary aerosol potential 25 times higher, 2.7 mg km-1.
- The secondary aerosol emission factors for a similar vehicle and driving cycle reported by (Karjalainen et al., (2016) and (Platt
   et al., (2013) are 4.3 mg km-1 and 12.7 mg km-1 (SOA only), respectively. The values are higher than in this study, possibly because the OH exposures are different: using a PAM reactor, (Tkacik et al., (2014) have shown that the SOA formation from

vehicle exhaust depends strongly on the OH exposure. Another reason for the higher values is probably that both Karjalainen et al. (2016) and Platt et al. (2013) used a cold-start cycle. In Karjalainen et al. (2016), most of the secondary mass is indeed formed in the beginning of the cycle, when the engine and the after-treatment system are cold. Interestingly, Karjalainen et al. (2016) do not observe a similar peak in the secondary mass formation at the end of the cycle as we see in Fig. 15c. However,

- 5 the gas measurements by Karjalainen et al. (2016) and Platt et al. (2013) show that during the last acceleration in the cycle, there are elevated concentrations of total hydrocarbons and ammonia, which could be potential sources of SOA and ammonium nitrate formation in an oxidation flow reactor.
- We also observe a new phenomenon, where engine braking results in high concentrations of secondary aerosol forming
  precursors. Every deceleration (i.e. engine braking) during warm NEDC produces a peak in secondary mass concentration. The tail at the beginning of the cycle is also a result from engine braking, as steady-state driving at 80 km h-1 was always performed before the warm NEDC. This phenomenon is not evident in the results of Karjalainen et al. (2016), since the mass concentration does not seem to correlate with vehicle speed in their study. However, they observe repeated events of nanoparticle growth in the PAM during the cycle, which could be related to engine braking. Because of the mixing of the sample inside the PAM, it is impossible to link these growth events to certain phases in the driving cycle.

sample inside the 1 Aw, it is impossible to hirk these growth events to certain phases in the driving cycle.

The time-resolved emission factor of secondary aerosol mass in Fig. 13 is achieved by multiplying the secondary mass concentration by the exhaust mass flow. Low exhaust mass flow during engine braking cancels out the high mass concentration peaks. Instead, the peak at the end of the cycle dominates the emissions of secondary aerosol precursors.

20

25

Since no aerosol chemical composition measurements were performed, we cannot specify the amount of organic mass in the formed secondary aerosol, and therefore we do not present the emission factor for SOA potential of the engine exhaust. In addition, the high background mass (i.e. unclean dilution air) and the lack of real-time  $OH_{exp}$  measurement make this data qualitative rather than quantitative. However, this experiment shows the feasibility of TSAR to measure the time-resolved secondary aerosol formation potential of rapidly changing vehicle emissions. This way, we can identify the driving conditions in which most secondary aerosol forming precursors are emitted. If the sample was injected to a smog chamber with a constant dilution ratio and then oxidized, like Platt et al. (2013) did, the precursor pulses emitted during engine braking events would cause an over-estimation of total secondary aerosol formation potential.

**4** Conclusions**

30 In this work, we introduced TSAR, a new short-residence-time oxidation flow reactor for secondary aerosol formation studiesmeasurements. We studied the performance of the reactor by measuring the sulfuric acid yield, toluene SOA yield as well as the composition and the secondary aerosol formation potential of light-duty gasoline vehicle exhaust during a transient driving cycle. In addition, we characterized the particle transmission efficiency and the residence time distribution of the reactor and did a modeling study on vapor losses in TSAR-

35

According to the model results, the vapor losses in TSAR are higher than in the reactors with longer residence times. The losses depend strongly on the condensational sink of the sample, which is usually high in exhaust measurements (resulting in lower losses). For applications with low condensational sink, we recommend longer residence time than in the TSAR or injection of seed aerosol. When there is no possibility for seed aerosol injection, a tradeoff must be made between fast response

40 and low vapor losses.

The toluene experiments show that both SOA yield and composition are similar in TSAR SOA as in PAM SOA, even though PAM operates in OFR185 mode and TSAR in OFR254 mode. The similarity indicates that the TSAR can be used instead of the OFR185 PAM reactor when high time-resolution is needed. is able to simulate atmospheric SOA formation since Bruns et al. (2015) and Lambe et al. (2015) show that the composition of SOA formed in PAM is similar to the SOA formed in a smog chamber.

5

10

The particle losses in TSAR are negligible and the flow is near-laminar. These properties, together with short residence time, make TSAR better suited for monitoring the secondary aerosol formation potential of rapidly changing emission sources than the PAM chamber. We demonstrate the importance of this feature by measuring the secondary aerosol formation of car exhaust during a driving cycle. This experiment shows that TSAR is able to differentiate which driving conditions are most significant

regarding the secondary aerosol formation potential.

**5** Data availability**

The data of this study is available from the authors upon request.

**6** Acknowledgements**

- The TSAR was designed and built in the project "Finnish-Chinese Green ICT R&D&I Living Lab for Energy Efficient, Clean 15 and Safe Environments", financially supported by Finnish Funding Agency for Innovation (Tekes), and Ahlstrom Oy, FIAC Invest Oy, Green Net Finland Oy, Kauriala Oy, Lassila & Tikanoja Oyj, Lifa Air Oy, MX Electrix Oy, Pegasor Oy and Sandbox Oy.
- The TSAR characterization was conducted in the framework of the HERE project funded by Tekes (the Finnish Funding 20 Agency for Innovation), Agco Power Oy, Dinex Ecocat Oy, Dekati Oy, Neste Oyj, Pegasor Oy and Wärtsilä Finland Oy.

Pauli Simonen acknowledges Tampere University of Technology Graduate School.

[revised manuscript text omitted]